# AGENTRX: Diagnosing AI Agent Failures from Execution Trajectories

## Abstract

AI agents often fail in ways that are difficult to localize because executions are probabilistic, long-horizon, multi-agent, and mediated by noisy tool outputs. We address this gap by manually annotating failed agent runs and release a novel benchmark of 115 failed trajectories spanning structured API workflows, incident management, and open-ended web/file tasks. Each trajectory is annotated with a critical failure step and a category from a grounded-theory derived, cross-domain failure taxonomy. To mitigate the human cost of failure attribution, we present AGENTRX, an *automated domain-agnostic* diagnostic framework that pinpoints the critical failure step in a failed agent trajectory. It synthesizes constraints, evaluates them step-by-step, and produces an auditable validation log of constraint violations with associated evidence; an LLM-based judge uses this log to localize the critical step and category. Our framework improves step localization and failure attribution over existing baselines across three domains.

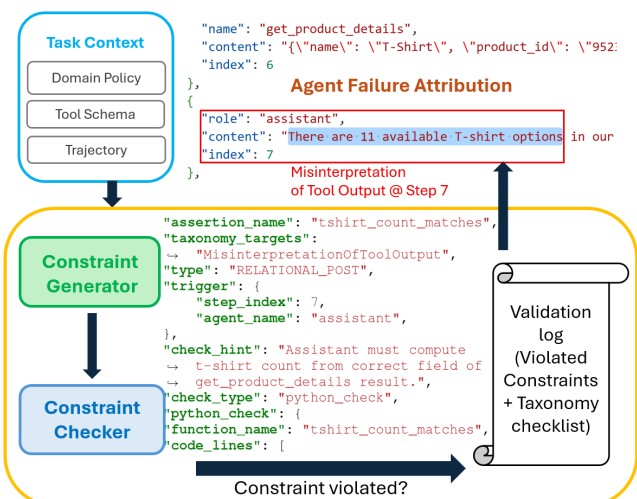

*Figure 1.* Given domain policy, tool schema, and a failed trajectory, AGENTRX outputs the critical failure step and a failure category.

## 1. Introduction

Large Language Models (LLMs) based agents are increasingly *acting autonomously* without a human in the loop. Modern AI agentic systems plan, coordinate, and execute complex and critical real-world tasks. These agents are no longer confined to sandboxes; they are being deployed in high-stakes environments such as recruiting (LinkedIn, 2024), web browsing (Fourney et al., 2024), and even operating production Cloud Services (Zhang et al., 2024). This ability to operate with minimal guidance dramatically amplifies human productivity Brynjolfsson et al. (2023). However, the same autonomy that enables speed also makes these agentic systems difficult to debug: trajectories are long-

horizon, tool outputs are noisy, and failures can propagate through side effects before they are observed. Consequently, ensuring the robustness of AI agents is not an option, but a prerequisite for their safe adoption in the real-world.

In this paper, our focus is on: *root-causing and localizing the critical failure which prevented the Agent from successful task completion.* We define *critical failure* as the first unrecoverable failure by any agent in an execution trajectory. To understand and characterize agentic failures, we manually analyze failed agent executions and introduce a benchmark of 115 agent execution trajectories across structured API workflows, incident management, and open-ended web/file tasks. Each trajectory includes the full execution trace (messages, tool calls, tool outputs, and observable environment state) and is annotated with the *critical step* and a *failure category* capturing why the run ultimately fails. The grounded-theory–based annotation process also yields a novel cross-domain failure taxonomy for root-cause attribution.

However, manual failure attribution is expensive and hard to scale. There is a significant need for automated validation of agents. In Software Engineering, reliability is enforced through contracts that make interface misuse and constraint violation explicit. Agentic systems require an analogous

[1]Anonymous Institution, Anonymous City, Anonymous Region, Anonymous Country. Correspondence to: Anonymous Author <anon.email@domain.com>.

Preliminary work. Under review by the International Conference on Machine Learning (ICML). Do not distribute.

notion of correctness, but stated over executions: ambiguous intents, long trajectories of messages, tool calls, observations, and state updates. To operationalize this view, we designed AGENTRX, a *domain-agnostic* automated framework for AI agents that (1) normalizes heterogeneous multi-agent logs into a common trajectory representation, (2) synthesizes *constraints* from tool schema, domain policies, and the observed trajectory (3) produces a step-indexed validation log of violated obligations with supporting evidence, which a downstream judge uses to localize the critical failure step and assign a category. As shown in Figure 1, after the assistant agent calls GET_PRODUCT_DETAILS and reports "11 available T-shirt options," AGENTRX synthesizes a relational constraint TSHIRT_COUNT_MATCHES that recomputes the count from the tool response and checks agreement with the assistant's interpretation. The resulting violation includes evidence (trajectory window), helping the judge localize the decisive failure.

We evaluate this framework on our proposed benchmark and prior work Zhang et al. (2025a). Our best method beats state of the art and baselines on both localizing the first unresolved critical failure step and categorizing the root-cause. Overall, the results show that trajectory-level constraints are a useful signal for detecting failures. Our contributions are: (i) An open-source dataset of **115 failed trajectories across 3 agentic domains**-including step-level annotations and failure categorization. (ii) a **novel domain-agnostic agentic failure diagnosis framework** that combines constraint synthesis and LLM-based adjudication using auditable violation logs. (iii) A **cross-domain failure taxonomy derived through grounded theory coding**, capturing 9 root-cause categories that generalize across diverse multi-agent settings, and (iv) Extensive experiments showing 23.6% absolute improvement in failure localization and 22.9% improvement in root-causing the failure.

## 2. The AGENTRX Benchmark

We are the first to introduce a benchmark for *attribution of the first unrecoverable failure* in AI agents. Our benchmark contains three diverse domains: API workflows, incident management, and real-world web/file tasks across both single-agent and multi-agent settings.For each domain, we sample the trajectories where an agent failed to solve a task and use them to construct our failure attribution benchmark. $\tau$-**bench.** $\tau$-bench Yao et al. (2024) is a tool-calling benchmark for retail tasks, with an LLM-simulated user and a single agent equipped with domain-specific APIs and policy guidelines. In a typical $\tau$-bench task, the agent can cancel or modify pending orders, return or exchange delivered orders, update user addresses, and answer product queries. We sample 115 $\tau$-bench trajectories using GPT-4O with one trial per task and analyze 29 trajectories that fail.

**Flash.** Flash Zhang et al. (2024) is a workflow automation agent designed for diagnosing recurring incidents in real-world production settings. It is a multi-agent system in which a team of specialized agents execute a troubleshooting guide for the incident being investigated. The guide involves steps like querying for clusters in a region that might be affected and also mitigation steps based on whether the incident is a false alarm or not. We sample 42 Flash trajectories containing at least one agent failure for our analysis.

**Magentic-One.** Magentic-One Fourney et al. (2024) is a generalist multi-agent system for open-ended web and file-based tasks. It comprises five specialized agents with distinct capabilities, including web browsing, file navigation, and code writing. We randomly sample 44 failed multi-agent trajectories from the Who&When dataset (Zhang et al., 2025a). Together, these sources give us a total of 115 failed trajectories containing at least one agent failure that form our AGENTRX benchmark.

**Comparison to Prior Work.** We qualitatively compare our annotation strategy with Who&When Zhang et al. (2025a) (W&W)'s labeling. Their annotated failure is not necessarily the critical one as per our definition. In one trajectory (Appendix B), the Orchestrator recovers from the failure marked by W&W: by switching from WebSurfer to FileSurfer, and only becomes unrecoverable after a later hallucinated file-/path error (which our annotation catches). Our annotation strategy catches this critical error. This indicates that **localization of the first unrecoverable critical failure is harder and more fine-grained**.

### 2.1. Grounded Theory Annotation of Agent Failures

To systematically analyze the critical agent failures, we manually annotate trajectories using a grounded theory coding process (Glaser & Strauss, 2017). Grounded Theory (GT) takes qualitative data and produces a theory in an iterative process. Unlike hypothesis-driven studies, GT develops hypotheses and a higher-level theory that emerge from the data. Instead of imposing a predefined label set for agent failures, we began with *open coding*, allowing categories to emerge from repeated failure patterns. Three annotators worked on each of the three domains and coded one trajectory at a time. During open coding, annotators read the trajectories step-by-step and wrote a structured *coding memo* whenever a failure was observed. We define a *failure* as any point at which the agent did not make progress toward a correct outcome *e.g.,* violating a policy guideline or executing an invalid tool call. We follow a two-stage coding procedure:

**Phase 1: Exhaustive failure marking.** Annotators first marked *all* failure steps, even if later steps dominate the final outcome. For each failure event, we recorded: (i) the step index; (ii) an *open code*: a short, concrete description

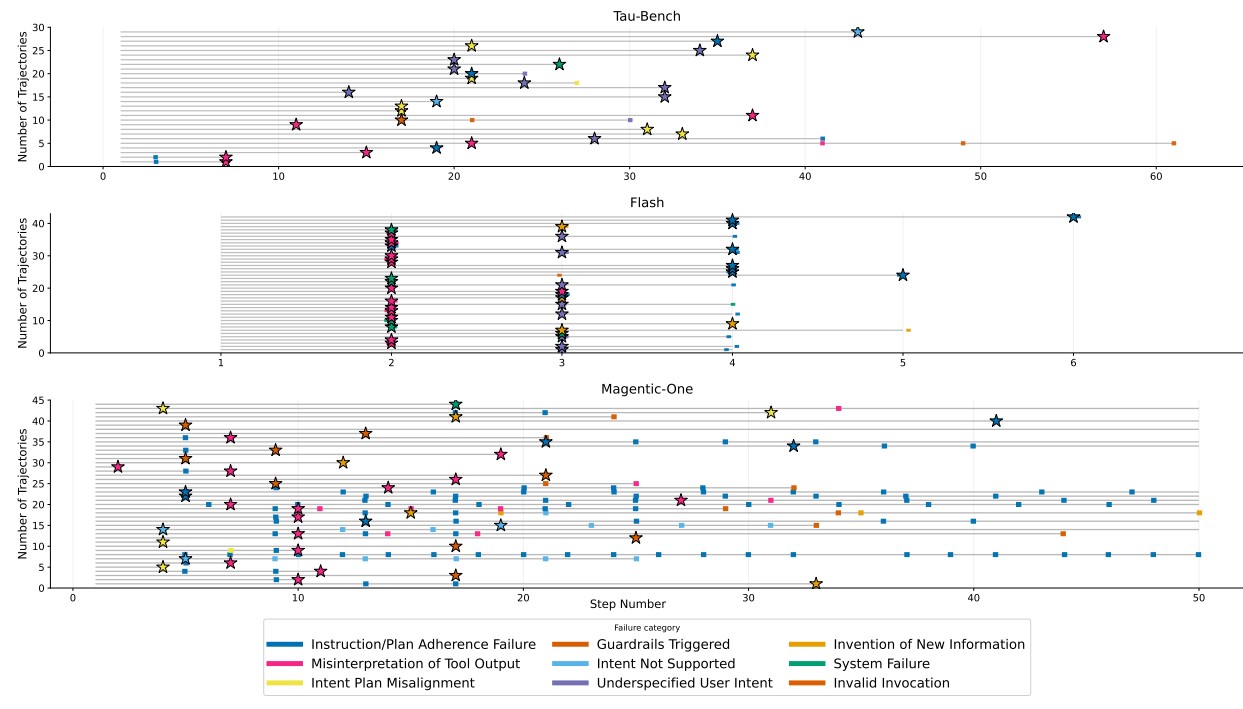

*Figure 2.* **Failure timelines per trajectory across domains.** Each horizontal line is one trajectory. Colored ticks mark detected failures at the corresponding step number (labeled with the category); the star indicates the root-cause step. Magentic-One has 295 total failures, with 68% of trajectories containing at least two failures; Flash has 52 total failures, and $\tau$-bench has 39.

tightly grounded in the trace; and (iii) a reason for that failure. This stage captures early deviations and cascading failures, including ones the agent later recovers from.

**Phase 2: Critical failure identification.** We then identified the *critical failure step* that leads to the incorrect terminal outcome. Concretely, we work backward from the terminal failure to the earliest failure from which the agent does not recover.

Throughout the coding process, new codes were constantly compared with previously assigned ones. When a new failure matched an existing category, the same code was reused; otherwise, we created a new code and wrote a concise definition. As coding progressed, we periodically aggregated related open codes into higher-level categories, discussed them extensively among annotators, and refined category definitions. We iterated on the taxonomy until we reached *theoretical saturation* where new trajectories did not introduce new failure phenomena. At that point, we *froze* the category definitions and used closed coding to annotate the remaining trajectories *and to recode the previously annotated ones*, verifying that the taxonomy covers failures beyond the subset that initially shaped it. Further, our labels are drawn from three structurally different domains spanning both single-agent and multi-agent settings, and each domain was annotated by independent annotators.

**Benchmark Format.** Each entry corresponds to a trajectory

and contains (i) an identifier, (ii) a record of failure events encountered during execution, and (iii) a critical failure. For each failure event, we store its step in the trajectory, a brief natural-language description, and a failure label from the taxonomy with supporting rationale. The list of recorded failures gives a causal chain from the first unrecoverable failure to the terminal one.

### 2.2. A Cross-Domain Failure Taxonomy

Prior work takes a system-level view of multi-agent failures, organizing failure modes by design, coordination, and verification stages (Pan et al., 2025). We develop a taxonomy that labels the critical failure:

1. **Plan Adherence Failure** The agent fails to follow required directions or the agreed plan by ignoring directives. This covers both under-execution (missed steps) and over-execution (unplanned actions, e.g., extra tool calls) that violate the plan, domain policy, or orchestrator constraints.

2. **Invention of new information** The agent introduces, removes, or alters information that is not grounded in any available input, context, or tool output. This includes fabricating unsupported facts, hallucinated details, or omitting relevant information without justification.

3. **Invalid Invocation** The agent issues an invalid tool call, e.g., malformed inputs, missing required arguments, or

*Table 1.* Distribution of *critical failure* labels across domains. We normalize per domain (percent of failed trajectories)

| Root-cause category | $\tau$-bench | Flash | Magentic |
|---|---|---|---|
| Instruction Adherence | 10.3 | 23.8 | 18.2 |
| Invention of Information | 0 | 9.5 | 9.1 |
| Invalid Invocation | 3.4 | 0 | 0 |
| Misinterpretation of Tool Output | 24.1 | 33.3 | 34.1 |
| Intent-Plan Misalignment | 24.1 | 0 | 9.1 |
| Under-specified Intent | 27.6 | 19 | 0 |
| Intent Not Supported | 6.9 | 0 | 6.8 |
| Guardrails Triggered | 0 | 0 | 20.5 |
| System Failure | 3.4 | 14.3 | 2.3 |
| **# failed trajectories** | **29** | **42** | **44** |

values that fail schema validation.

4. **Misinterpretation of Tool Output** The agent incorrectly reasons about its own or another agent's tool output, leading to incorrect assumptions or actions.

5. **Intent–Plan Misalignment** The agent misinterprets the user's goal or constraints and produces an incorrect plan.

6. **Under-specified User Intent** The agent was unable to complete the task due to lack of complete information at that point in the trajectory.

7. **Intent Not Supported** The agent is asked to perform an action for which a tool is not available.

8. **Guardrails Triggered** The agent's execution is blocked by safety policies or external access restrictions.

9. **System Failure** The agent faces a system connectivity issue while calling a particular tool like an API endpoint not being reachable.

**Annotation Cost.** The manual effort required for these annotations is substantial. Annotators spent, on average, 20 minutes per $\tau$-bench trajectory, 22 minutes per Flash trajectory, and 24 minutes per Magentic trajectory. Across 115 trajectories, this amounts to 42.7 total human hours and is challenging to scale. This motivates the need for automated failure attribution frameworks such as AGENTRX.

## 3. AgentRx Framework Formulation

**Setup.** AGENTRX takes as input (i) a toolset $\mathcal{T}$ (available tools/agents with input–output schema), (ii) an optional domain policy $\Pi$ (domain-specific natural-language rules), and (iii) a failed trajectory $T = \langle s_1, \ldots, s_n \rangle$. Each step $s_k$ contains multiple logged events with fields such as agent name, tool name, step index, tool output or natural language conversational content. We normalize each trajectory into a common intermediate representation (IR) because formats differ across domains. Let $T_{\leq k}$ denote the trajectory prefix up to and including step $k$. At a high level, AGENTRX generates *executable constraints* from the tool schema, domain

policy, and $T_{\leq k}$, and evaluates them step-by-step to produce an auditable validation log V of violations and supporting evidence. This log is then passed to an LLM-based judge along with a taxonomy checklist K to predict a root-cause step $\hat{s}$ and failure category $\hat{y}$.

**Taxonomy checklist.** We compile our failure taxonomy into a semantic checklist K: for each category $y$ in the taxonomy, $K(y)$ specifies a small set of targeted yes/no questions (and brief decision criteria) that operationalize the category definition.

### 3.1. Global and Dynamic Constraint Synthesis

AGENTRX synthesizes constraints in two stages: *Global constraints* $\mathcal{C}^G$ are synthesized once from the tool schema and domain policy, and stored in a global store G:

$$\mathcal{C}^G \leftarrow \text{SYNGLOBAL}(\mathcal{T}, \Pi)$$

*Dynamic constraints* $\mathcal{C}_k^D$ are synthesized at each step from the task instruction $I$ and the observed prefix $T_{\leq k}$, using G as context, and stored in a local store $L_k$:

$$\mathcal{C}_k^D \leftarrow \text{SYNDYNAMIC}(\mathcal{T}, I, T_{\leq k}, \text{G})$$

Intuitively, global constraints encode domain rules, while dynamic constraints encode trajectory-specific rules and prior observations (e.g., constraints that must hold given earlier tool outputs). We define the constraints available at step $k$ as the union $\mathcal{C}_k := \mathcal{C}^G \cup \mathcal{C}_k^D$. In practice, constraint synthesis emits constraints in a lightweight schema to standardize checking across domains (subsection E.1).

### 3.2. Guarded Constraints and Evaluation

A constraint $C$ generated by AGENTRX comprises (i) a guard $G_C(T_{\leq k}, s_k) \in \{0, 1\}$ that determines when the constraint applies and (ii) an assertion $\Phi_C(T_{\leq k}, s_k) \in \{\text{SAT}, \text{VIOL}\}$ that returns a binary verdict $v$ when evaluated. Guards typically capture structural conditions (e.g., "this step contains a call to tool $t$", or "a specific agent is invoked"). Evaluation at step $k$ is:

$$\text{EVAL}_C(k) := \text{EVAL}(C, T_{\leq k}, s_k)$$

$$\text{EVAL}_C(k) = \begin{cases} (\text{SKIP}, \emptyset) & \text{if } G_C(T_{\leq k}, s_k) = 0 \\ (\Phi_C(T_{\leq k}, s_k), e) & \text{otherwise.} \end{cases}$$

Constraints are enforced either with programmatic checks or semantic checks. Programmatic checks are predicates over structured fields (e.g., schema validity, equality, membership), while semantic checks are natural-language predicates evaluated using an LLM-based checker. Both types of checks return a verdict–evidence pair $(v, e)$. Examples in Appendix E.

### 3.3. Validation Log

At step $k$, AGENTRX evaluates each constraint $C \in \mathcal{C}_k$ whose guard applies, i.e., $G_C(T_{\leq k}, s_k) = 1$. We record each violated constraint and its supporting evidence in a step-indexed validation log:

$$\mathsf{V} := \{(k, C, e) \mid C \in \mathcal{C}_k, \ G_C(T_{\leq k}, s_k),$$
$$(\text{VIOL}, e) = \text{EVAL}(C, T_{\leq k}, s_k)\}.$$

$\mathsf{V}$ is step-indexed, auditable, and directly links each violation to supporting evidence.

### 3.4. Judge for Root Cause Attribution

Given the task instruction $I$, trajectory $T$, checklist $\mathsf{K}$ and violation log $\mathsf{V}$, an LLM-based judge outputs: (i) a critical failure step $\hat{s}$, and (ii) a critical failure category $\hat{y}$ from our taxonomy.

**Critical step selection.** The judge selects $\hat{s}$ as the first step $k$ whose violation(s) are sufficient to explain why the run fails with respect to $I$ (i.e., correcting step $k$ would plausibly change the outcome). Violations serve as diagnostic evidence rather than hard requirements; the judge may override them when the trajectory context indicates otherwise.

**Failure category assignment.** Each violation is mapped to multiple possible taxonomy labels (e.g., invalid tool invocation or misinterpretation of tool output). The judge sets $\hat{y}$ based on the taxonomy labels associated with the violation(s), and emits a short rationale citing the specific reason for selecting the failure category.

## 4. Experimental Evaluation

AGENTRX outputs (i) a *critical step index* $\hat{s}$ and (ii) a *failure category* $\hat{y}$ from a fixed cross-domain taxonomy. We evaluate AGENTRX along two axes: **(i) accuracy** of step localization and category attribution, and **(ii) robustness** to cross-domain shift, different constraint generation strategies, judge stochasticity, and long traces.

**Benchmark.** We evaluate on our AGENTRX benchmark across three domains: (i) API workflows ($\tau$-bench), (ii) incident response workflows (Flash), and (iii) open-ended web/file tasks (Magentic-One). Each instance is a step-indexed trajectory containing user/agent messages, tool invocations, tool outputs, and any logged environment state available to the agent. We report per-domain statistics: $\tau$-bench has 29 trajectories with median length 36 (range 20–62), Flash has 42 with median length 3 (2–6), and Magentic-One has 44 with median 33 (5–130 range). Flash also averages 8 substeps per step.[1]

---

[1]Flash steps are defined using the Troubleshooting Guide for consistency; each step expands into multiple substeps.

### 4.1. Evaluation Metrics and Experimental Settings

**Step Localization Accuracy.** We treat step localization as a single-step identification problem over the executed trajectory. We report:

(i) **Critical Step-index Accuracy**, the fraction of trajectories where the predicted step exactly matches the annotated step;

(ii) **Critical Step-index Accuracy@$\pm r$**, the fraction of trajectories where the predicted step falls within $\pm r$ steps of the annotated step, for $r \in \{1, 3, 5\}$; and

(iii) **Average Step Distance** (lower is better), which measures how far predictions are from the annotated step. Together, these metrics capture both strict correctness and utility when diagnoses are off by a small number of steps.

**Failure Category Accuracy.** This involves assigning the failure to a category in our taxonomy, or outputting *inconclusive* (or a custom label). We report:

(i) **Critical Failure Category Accuracy**, where the predicted failure category matches the annotated category;

(ii) **Any-failure Category Accuracy**, where the prediction matches *any* category among the trajectory's failure steps;

(iii) **Earliest Category Accuracy**, where it matches the category at the first failure step; and

(iv) **Terminal Category Accuracy**, where it matches the category at the last failure step.

**Experimental settings.** We run each configuration $n=3$ times and report mean $\pm$ standard deviation for step-index accuracy, failure category accuracy, and average step distance (Table 5). This quantifies robustness to sampling noise and judge variability. We enable both (i) *programmatic constraints* and (ii) *semantic constraints* (natural-language rules). Table 5 reports results with both enabled to reflect the default AGENTRX setting. We use GPT-5 as our default model for all experiments. (Results using o3 and without semantic constraints for $\tau$-bench in Appendix A).

### 4.2. Critical Failure Localization in Agentic Systems

We compare our baseline against Who&When (W&W) on $\tau$-bench and Magentic-One. For $\tau$-bench, we evaluate on all 29 trajectories; for Magentic-One, we evaluate on the subset of 16 (out of the 44) trajectories where our dataset has the same annotated step number as W&W. We run W&W's LLM-as-a-judge, with one prompt modification: instead of returning the first failure step, we ask it to identify the *first unrecoverable* critical step. Even with this modification, our simplest baseline achieves higher step and agent accuracy than the best performing variant of W&W on $\tau$-bench and Magentic, as shown in Table 2. This improvement comes

*Table 2.* **Failure attribution accuracy.** Agent accuracy identifies the failed agent; step accuracy identifies the failure step. We run both methods with GPT-5; W&W* denotes our prompt-modified version. We report W&W's best-performing variant.

| Method | $\tau$-bench (# of traj = 29) | | **Magentic** (# of traj = 16) | |
| --- | --- | --- | --- | --- |
| | Agent (%) | Step (%) | Agent (%) | Step (%) |
| Who&When* | 62 | 17.2 | 6.2 | 56.3 |
| Our Baseline | 75.9 | 32.2 | 81.2 | 56.3 |

*Table 3.* Mean token and characters per trajectory and per step.

| Metric | $\tau$-Bench | Flash | Magentic |
| --- | --- | --- | --- |
| **Avg tokens / step** | 133 | 169 | 330 |
| **Avg chars / step** | 480 | 930 | 1280 |
| **Avg tokens / trajectory** | 4889 | 6415 | 16484 |

from the *strength of our baseline*: **it is grounded in our comprehensive failure taxonomy and prompts the LLM to use the same failure attribution procedure as our human annotators** (subsection G.1).

In addition, W&W's reported magentic step accuracy uses their step-by-step variant, which requires $16\times$ more LLM calls per trajectory than all-at-once and is therefore more expensive. Their all-at-once variant, which is closest to our baseline, achieves only 12.5% step accuracy on Magentic.

### 4.3. Overall Failure Attribution Performance

We compare our baseline with the best-performing version of AGENTRX. **AGENTRX beats baseline in all three domains in both critical step-index and failure category accuracy.** Notably, in $\tau$-bench the step accuracy jumps from 32.2% in baseline to 54% with the best performing variant as shown in Table 5. Even category accuracy improves from 25.3% to 40.2%. For Flash, where our baseline is already strong, the category accuracy improves by 6.4 points (53.9% $\rightarrow$ 60.3%), while the step accuracy improves by 2.4 points. Magentic is a difficult subset of our benchmark with an average of 6.7 failures per trajectory. Our best configuration of AGENTRX beats baseline by a small margin. We also report results on Magentic*, a filtered subset of Magentic with 27 trajectories of length at most 50 steps to keep trajectories within a manageable horizon. On Magentic*, AGENTRX improves baseline step accuracy from 42% to 46.9% (and category accuracy from 39.5% to 44.4%). Notably, the average step distance also reduces in AGENTRX compared to baseline for all three domains.

### 4.4. Impact of Violations and Taxonomy Checklist

We consider three judge inputs: (i) violation evidence alone (Baseline+Vio.), (ii) checklist alone (Taxonomy Checklist), and (iii) checklist+violations (Checklist+Vio.). On $\tau$-bench,

*Table 4.* We ablate AGENTRX by enabling only global constraints or only dynamic constraints for $\tau$-bench. Mean $\pm$ std over $n=3$.

| Method | Step-index acc. (%) $\uparrow$ | Category acc. (%) $\uparrow$ |
| --- | --- | --- |
| Baseline | $32.2 \pm 3.2$ | $25.3 \pm 1.6$ |
| Global-Only | $41.4 \pm 2.8$ | $28.7 \pm 1.6$ |
| Dynamic-Only | $43.7 \pm 1.6$ | $36.8 \pm 1.6$ |
| AGENTRX | $48.3 \pm 0$ | $39.1 \pm 1.6$ |

violations alone yield a large gain over the baseline in step accuracy (32.2$\rightarrow$47.1) and category accuracy (25.3$\rightarrow$37.9), while the checklist alone shows no gains. On Flash, both signals help: the checklist alone improves category accuracy (53.9$\rightarrow$57.9), while checklist plus violations further increases it to 60.3; step accuracy also improves, from 80.9 to 83.3 using checklist alone (Table 5). On Magentic, the strongest configuration uses the checklist alone, suggesting that **semantic structure can outweigh sparse or noisy violations evidence.** The checklist plus violations setting outperforms baseline on all three metrics in Magentic*. In most settings, checklist plus violations improves over either signal alone. **Overall, the validation log of violations and taxonomy checklist provide useful signal for critical failure attribution.**

### 4.5. Impact of Constraint Generation Strategies

We evaluate two ways of generating constraints from the tool schema and optional domain policy when diagnosing a trajectory: *step-by-step* generation, which conditions on the trajectory prefix, and *one-shot* generation, which conditions on the full trajectory. In the step-by-step setting, at each step $k$ we generate constraints conditioned only on the prefix $T_{\leq k}$ (i.e., all events observed up to and including step $k$). This yields *step-conditioned* constraints that enable incremental checking and closer to a real-world setting. In the one-shot setting, we provide the entire trajectory to the generator and produce a single set of constraints in one pass. **One-shot constraints are generated at once over the entire trajectory, serving as a cost-efficient alternative.**

Table 5 compares one-shot vs. step-by-step constraint generation across domains.

On $\tau$-bench, one-shot performs better: the best one-shot setting reaches 54.0% step accuracy and 40.2% category accuracy, whereas the best step-by-step variant reaches 41.4% and 35.6%, respectively. Flash shows a mixed pattern: step-by-step improves category accuracy, while step-index accuracy is similar across the two settings. In Magentic, longer trajectories make one-shot constraint generation brittle, we do not see a huge improvement over the baseline. We evaluated the step-by-step setting only on Magentic* because of long-horizon trajectories. Step-by-step yields a sizeable gain in both step (46.9% vs. 42%) and category accuracy

*Table 5.* **Effect of judge inputs and constraint generation.** We compare one-shot vs. step-by-step constraint generation, judge inputs (violations, taxonomy checklist, both) and judging protocols (Step-then-Category or All-at-Once).

| Metric ($n=3$) | Judge Baselines | | AGENTRx Ablations | | | |
|---|---|---|---|---|---|---|
| | Baseline | Step-then-Cat. | Baseline+Vio. | Step-then-Cat.+Vio. | Taxonomy Checklist | Checklist+Vio. |
| **Tau-Bench** | | | One-shot Constraint Generation | | | |
| Step-index acc. (%) ↑ | $32.2 \pm 3.2$ | $32.2 \pm 1.6$ | $47.1 \pm 1.6$ | $\boxed{54 \pm 1.6}$ | $32.2 \pm 1.6$ | $48.3$ |
| Category acc. (%) ↑ | $25.3 \pm 1.6$ | $27.6 \pm 2.8$ | $37.9 \pm 2.8$ | $\boxed{40.2 \pm 1.6}$ | $25.3 \pm 1.6$ | $39.1 \pm 1.6$ |
| Avg. step distance ↓ | $5.7 \pm 0.8$ | $6 \pm 1$ | $2.8 \pm 0.3$ | $\boxed{2.4 \pm 0.5}$ | $5.8 \pm 0.7$ | $3 \pm 0.3$ |
| **Tau-Bench** | | | Step-by-Step Constraint Generation | | | |
| Step-index acc. (%) ↑ | $32.2 \pm 3.2$ | $32.2 \pm 1.6$ | $41.4 \pm 2.8$ | $36.8 \pm 1.6$ | $32.2 \pm 1.6$ | $37.9 \pm 2.8$ |
| Category acc. (%) ↑ | $25.3 \pm 1.6$ | $27.6 \pm 2.8$ | $35.6 \pm 1.6$ | $34.5$ | $25.3 \pm 1.6$ | $35.6 \pm 1.6$ |
| Avg. step distance ↓ | $5.7 \pm 0.8$ | $6.0 \pm 1$ | $3.3 \pm 0.6$ | $4.1 \pm 0.1$ | $5.8 \pm 0.7$ | $3.6 \pm 0.1$ |
| **Flash** | | | One-shot Constraint Generation | | | |
| Step-index acc. (%) ↑ | $80.9 \pm 2.3$ | $73 \pm 1.3$ | $81.8 \pm 1.3$ | $70.6 \pm 2.3$ | $\boxed{83.3 \pm 2.3}$ | $80.1 \pm 1.3$ |
| Category acc. (%) ↑ | $53.9 \pm 3.6$ | $55.5 \pm 3.6$ | $53.9 \pm 7.6$ | $52.3 \pm 2.3$ | $57.9 \pm 2.7$ | $58 \pm 3.6$ |
| Avg. step distance ↓ | $0.25 \pm 0.2$ | $0.34 \pm 1.3$ | $\boxed{0.2}$ | $0.36$ | $0.2 \pm 0.3$ | $0.2$ |
| **Flash** | | | Step-by-Step Constraint Generation | | | |
| Step-index acc. (%) ↑ | $80.9 \pm 2.3$ | $73 \pm 1.3$ | $76.2$ | $68.2 \pm 5.4$ | $83.3 \pm 2.3$ | $76.1 \pm 2.3$ |
| Category acc. (%) ↑ | $53.9 \pm 3.6$ | $55.5 \pm 3.6$ | $57.1 \pm 6.2$ | $59.5 \pm 2.3$ | $57.9 \pm 2.7$ | $\boxed{60.3 \pm 1.3}$ |
| Avg. step distance ↓ | $0.25 \pm 0.2$ | $0.34 \pm 1.3$ | $0.3$ | $0.4$ | $0.2 \pm 0.3$ | $0.3$ |
| **Magentic** | | | One-Shot Constraint Generation | | | |
| Step-index acc. (%) ↑ | $31.8$ | $29.5 \pm 2.3$ | $25 \pm 2.3$ | $27.3 \pm 1.3$ | $\boxed{31.8}$ | $24.2 \pm 1.3$ |
| Category acc. (%) ↑ | $36.4 \pm 3.6$ | $37.8 \pm 5.7$ | $31.1 \pm 3.5$ | $34.1 \pm 4.5$ | $\boxed{37.1 \pm 5.7}$ | $25 \pm 1.3$ |
| Avg. step distance ↓ | $22 \pm 1$ | $13.4 \pm 2$ | $28 \pm 1.2$ | $\boxed{13.7 \pm 2.3}$ | $25.5 \pm 1.2$ | $28.3 \pm 1$ |
| **Magentic*** | | | Step-by-Step Constraint Generation | | | |
| Step-index acc. (%) ↑ | $42 \pm 1.8$ | $40.7$ | $45.7 \pm 1.8$ | $40.7 \pm 5.2$ | $42 \pm 1.8$ | $\boxed{46.9 \pm 3.5}$ |
| Category acc. (%) ↑ | $39.5 \pm 3.5$ | $40.7 \pm 5.2$ | $43.2 \pm 1.8$ | $42 \pm 1.8$ | $35.8 \pm 1.8$ | $\boxed{44.4 \pm 3}$ |
| Avg. step distance ↓ | $5 \pm 0.7$ | $4.9 \pm 0.8$ | $4.9 \pm 0.9$ | $5.2 \pm 0.4$ | $6.6 \pm 0.3$ | $\boxed{4.8 \pm 0.8}$ |

(39.5% vs. 44.4%) for Magentic*. These results align with the trajectory length across domains (Table 3). $\tau$-bench averages 4,889 tokens per trajectory, so one-shot generation can effectively condition on relevant context in a single pass. Magentic averages 16,484 tokens, where one-shot is more susceptible to context dilution and localized step-by-step constraint generation is more reliable.

We also distinguish between (i) **global constraints** derived solely from the tool schema and domain policy (trajectory-independent), and (ii) **dynamic constraints** generated conditionally from the observed prefix $T_{\leq k}$ (trajectory-dependent). We evaluate on these two ablations: **Global-Only** (schema/policy only) and **Dynamic-Only** (prefix-conditioned only). We run this experiment only on $\tau$-bench because Flash and Magentic do not have a domain policy. In Table 4, both ablations beat the baseline, showing that **either global constraints or dynamic constraints alone provide useful signal** on $\tau$-bench. Dynamic-Only is the stronger single component, especially for category accuracy but **combining both yields the best performance overall.**

### 4.6. LLM-as-a-Judge Evaluation

We evaluate failure attribution with an LLM-as-judge given the trajectory, violation evidence and taxonomy checklist. We consider two judging protocols. In *All-at-Once*, the judge sees the full validation log (and optionally the taxonomy checklist) in a single call and predicts both the critical step and failure category. In *Step-then-Category*, the judge first selects the failure step and then, conditioned on that step, predicts the category in a second call. The default AGENTRx setting is All-at-Once. Step-then-Category can be brittle: it *commits* to a single step before considering the failure taxonomy semantics, so a noisy step prediction can cascade into an incorrect label. This is visible on Flash, where Step-then-Category reduces step accuracy ($80.9 \rightarrow 73.0$) and is further reduced with noisy violations. Magentic and Magentic* follow a similar trend. In contrast, $\tau$-bench shows the opposite trend: Step-then-Cat.+Vio. is the best setting (54.0 step-index, 40.2 category) possibly due to more compact trajectories. *Step-then-Category can show substantial improvements unless the trajectories are too long.*

*Table 6.* **Step and category accuracy under tolerance** Mean $\pm$ std over $n=3$ runs.

| Domain / Method | Step Acc. ↑ | Acc@ ±1 ↑ | Acc@ ±3 ↑ | Acc@ ±5 ↑ | Avg. Dist. ↓ | Critical Cat. ↑ | Any Cat. ↑ | Earliest Cat. ↑ | Terminal Cat. ↑ |
|---|---|---|---|---|---|---|---|---|---|
| $\tau$-Bench (Baseline) | $32.2 \pm 3.2$ | $36.8 \pm 1.6$ | $50.6 \pm 1.6$ | $66.7 \pm 1.6$ | $5.7 \pm 0.8$ | $25.3 \pm 1.6$ | $35.6 \pm 1.6$ | $31 \pm 0$ | $23 \pm 3.2$ |
| $\tau$-Bench (AGENTRX) | $54 \pm 1.6$ | $59.8 \pm 1.6$ | $72.4 \pm 2.8$ | $83.9 \pm 1.6$ | $2.4 \pm 0.5$ | $40.2 \pm 1.6$ | $41.4 \pm 2.8$ | $35.6 \pm 0$ | $34.5 \pm 2.8$ |
| Flash (Baseline) | $80.9 \pm 1.9$ | $94.4 \pm 1.1$ | $100$ | $100$ | $0.2$ | $53.9 \pm 3$ | $62.7 \pm 4.4$ | $53.2 \pm 2.9$ | $62.7 \pm 4.4$ |
| Flash (AGENTRX) | $83.3 \pm 1.9$ | $98.4 \pm 1.1$ | $100$ | $100$ | $0.2$ | $60.3 \pm 1.1$ | $65.8 \pm 2.2$ | $58 \pm 1.1$ | $65.8 \pm 2.2$ |
| Magentic (Baseline) | $31.8 \pm 1.8$ | $40.9$ | $50 \pm 3.7$ | $53.3 \pm 2.1$ | $22.1 \pm 8.4$ | $36.3 \pm 1.8$ | $58.3 \pm 1$ | $34.1 \pm 1.8$ | $40.1 \pm 1$ |
| Magentic (AGENTRX) | $31.8 \pm 1.8$ | $40.9 \pm 3.7$ | $47.7 \pm 3.7$ | $50.8 \pm 1$ | $25.5 \pm 1$ | $37.1 \pm 4.6$ | $57.6 \pm 5.9$ | $32.6 \pm 2.1$ | $41.7 \pm 2.8$ |
| Magentic* (Baseline) | $42 \pm 1.8$ | $60.5 \pm 3.5$ | $69.1 \pm 4.6$ | $69.1 \pm 4.6$ | $5 \pm 0.7$ | $39.5 \pm 3.5$ | $60.5 \pm 4.6$ | $39.5 \pm 4.6$ | $46.9 \pm 4.6$ |
| Magentic* (AGENTRX) | $46.9 \pm 3.5$ | $61.7 \pm 6.3$ | $72.8 \pm 4.6$ | $79 \pm 3.5$ | $4.8 \pm 0.8$ | $44.4 \pm 3$ | $67.9 \pm 3.5$ | $42 \pm 1.8$ | $44.4 \pm 3$ |

## 4.7. Step & Category Accuracy Under Different Settings

**Step accuracy increases as we allow more tolerance in the predicted step** (Acc@±1 < Acc@±3 < Acc@±5) as shown in Table 6. We also report multiple category accuracy metrics because the right notion of correctness depends on use: critical failure accuracy measures whether the key failure is identified, while Any-failure is a weaker but often practical signal to flag some real failure even if the precise cause is missed. These results indicate that **our framework can be effectively used for agent failure attribution across a range of scenarios.**

## 5. Related Work

**Benchmarks for Agent Evaluation.** Recent work has introduced benchmarks that cover tool execution, web interaction, and assistant behavior (Barres et al., 2025; Yao et al., 2022; Drouin et al., 2024). AgentBench Liu et al. (2023) evaluates LLM agents across eight interactive environments that span code-centric, game-based, and web-based settings, including domains like operating systems and databases. WebArena Zhou et al. (2023) evaluates long-horizon web agents on real websites, testing whether they can convert language instructions into executable browser interactions. GAIA Mialon et al. (2023) targets general assistants with problems that are easy for humans but often require multi-step tool use, and has been used to evaluate generalist agent systems. However, these benchmarks primarily report terminal success and do not include gold failure-attribution annotations. Our dataset annotates the first unrecoverable critical step and assigns a failure category from a cross-domain taxonomy.

**Improving Reliability of Agents.** Koohestani (2025) provides runtime assurance for AI agents by monitoring executions against learned behavioral models. Recent work also translates plans into formal representations and applies model checking to validate alignment between an agent's behavior and its plan (Ramani et al., 2025; Zhang et al., 2025b). These methods primarily target runtime enforce-

ment or plan-conformance checks; our focus is to diagnose trajectories by extracting constraints from tools, policies, and the observed prefix, and use the resulting violations as evidence for failure attribution. A separate line of work improves agent performance through feedback and self-correction. Self-Refine (Madaan et al., 2023) formalizes an iterative loop in which an LLM generates critiques of its own outputs and applies revisions, showing gains across diverse tasks without additional training. Critic (Gou et al., 2024) uses external tools like search and code execution as feedback to verify and repair model outputs. Feedback is also used at training time to improve agents. Reflection and reinforced self-training methods, such as ReReST (Dou et al., 2025), iteratively curate better trajectories using reflection and reward-based filtering.

**LLM Evaluation of Agent Trajectories.** LLM-as-a-Judge uses state-of-the-art LLMs to score outputs against rubrics, reducing the cost of human evaluation. Gu et al. (2025) systematizes judge design and mitigation strategies for inconsistency and bias. Agent-as-a-Judge Zhuge et al. (2024) uses tool-using agents to evaluate other agents, arguing benefits over single-shot judges. AGENTRX is compatible with these paradigms, but shifts the judge from scoring to diagnosis by providing a step-indexed violation log as evidence.

## 6. Conclusion and Limitations

Our failure taxonomy covers three diverse domains and annotations from three independent annotators. However, it may not cover all failure modes in other agentic domains and may require extension. Sometimes AGENTRX can be misled when validation signals are weak or contain false positives. This motivates future work on identifying the smallest set of high-quality signals needed to separate true failures from noisy flags. Although AGENTRX relies on LLM calls, it reduces manual effort for failure attribution and is easier to scale. AGENTRX also offers cost-efficient one-shot constraint generation as an alternative. We view both the framework and the dataset as a step toward systematic, evidence-based failure attribution in AI agents.

## Impact Statement

This paper is focused on making AI agents easier to debug. As agents increasingly run multi-step workflows: calling tools, changing state, and coordinating sub-agents failures are difficult. They can be wrong tool calls, skipped confirmations, misread outputs, or decisions that push the run onto a path it never recovers from. Our goal is to help developers pinpoint where the run first became unrecoverable and why, using evidence already present in execution traces.

The main upside is practical: faster debugging, clearer accountability, and safer iteration. A system that can surface the root cause and connect it to trace evidence can make it easier to measure reliability. The benchmark and taxonomy also encourage better instrumentation and more careful thinking about what should be logged and what should be checked.

There are also risks. Better diagnosis does not automatically mean agents are safe to deploy, and explanations can be over-trusted if people treat them as guarantees. Trajectory data can also contain sensitive information depending on the application, so anyone using these methods should handle logs carefully and follow appropriate privacy practices. Finally, a taxonomy learned from existing benchmarks can inherit their blind spots, so it should be treated as a starting point rather than a complete map of real-world failures. Overall, we expect this work to be most useful as an Agentic debugging framework: it helps find problems earlier and fix them proactively, but it does not replace careful testing.

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

## A. Additional Experiments

*Table 7.* We evaluate AGENTRX by adding checklist and synthetic few-shot examples for $\tau$-bench (one-shot). Mean $\pm$ std over $n=3$.

| Method | Step-index acc. (%) ↑ | Category acc. (%) ↑ |
|---|---|---|
| Baseline | $32.2 \pm 3.2$ | $25.3 \pm 1.6$ |
| Baseline + Viol. | $47.1 \pm 1.6$ | $37.9 \pm 2.8$ |
| Examples | $33.3 \pm 1.6$ | $31.0 \pm 2.8$ |
| Examples + Viol. | $49.4 \pm 1.6$ | $46.0 \pm 1.6$ |

*Table 8.* We ablate AGENTRX by enabling and disabling natural language (NL) checks during constraint generation for $\tau$-bench (step-by-step). Mean $\pm$ std over $n=3$.

| Method | Step-index acc. (%) ↑ | Category acc. (%) ↑ |
|---|---|---|
| Baseline | $32.2 \pm 3.2$ | $25.3 \pm 1.6$ |
| Without NL Check Viol. | $36.8 \pm 1.6$ | $32.2 \pm 3.2$ |
| With NL Check Viol. | $41.4 \pm 2.8$ | $35.6 \pm 1.6$ |

*Table 9.* **o3 judge ablation for $\tau$-bench.** We compare evidence sources for root-cause attribution: violations-only, violations+taxonomy-checklist, checklist-only, and the baseline judge. Metrics: step-index accuracy and root-cause category accuracy (mean $\pm$ std over $n=3$ runs).

| Metric | Baseline | Viol. | Chk. | Chk.+Viol. |
|---|---|---|---|---|
| Step Acc. ↑ | $32.2 \pm 3$ | $37.9 \pm 2$ | $33.3 \pm 1.6$ | $41.4 \pm 3$ |
| Cat. Acc. ↑ | $25.3 \pm 2$ | $34.5 \pm 3$ | $29.9 \pm 4.3$ | $35.6 \pm 2$ |

## B. Root-Cause Attribution VS First Failure

The following trajectory demonstrates why root-cause failure attribution is the better objective for evaluating agent reliability. The Who&When dataset labels step 3 as failure citing that "WebSurfer's inability to reliably access the requested documents resulted in the overall task failure, as the necessary time span data could not be extracted or compared. This underscores the need for enhanced fallback mechanisms and more robust search strategies." This diagnosis misses a key fact: the agent recovers from this early access failure, so it is not the cause of the eventual outcome. We annotate the key failure as shown below:

```
1  {
2  "trajectory_id": "5f982798-16b9-4051-ab57↵
   ↪ -cfc7ebdb2a91",
3  "failures": [
4  {
5  "failure_id": 1,
6  "step_number": 13,
7  "step_reason": "Websurfer could not
   ↪ download a PDF file and search
   ↪ throught it which was an
   ↪ instruction given by
   ↪ Orchestrator.",
```

```
550         "failure_category": "Instruction/Plan
551     ↪    Adherence Failure",
552         "category_reason": "Instruction not
553     ↪    followed, the agent did not
554     ↪    download and search through the
555     ↪    PDF file as instructed",
556         "failed_agent": "Websurfer"
557     },
558     {
559
560         "failure_id": 2,
561         "step_number": 17,
562         "step_reason": "Websurfer could not
563     ↪    download a PDF file and search
564     ↪    throught it which was an
565     ↪    instruction given by
566     ↪    Orchestrator",
567         "failure_category": "Instruction/Plan
568     ↪    Adherence Failure",
569         "category_reason": "Websurfer could
570     ↪    not  download a PDF file and
571     ↪    search throught it which was an
572     ↪    instruction given by
573     ↪    Orchestrator",
574         "failed_agent": "Websurfer"
575     },
576     {
577
578         "failure_id": 3,
579         "step_number": 33,
580         "step_reason": "FileSurfer
581     ↪    hallucinated. It downloaded the
582     ↪    file at '/workspace/workspace/htt⌋
583     ↪    p:/export.arxiv.org/pdf/2007.xx'
584     ↪    but later attempted to read a
585     ↪    non-existent file
586     ↪    'file:///workspace/path_to_july_2⌋
587     ↪    020_paper.pdf'.",
588         "failure_category": "Invention of new
589     ↪    information",
590         "category_reason": "FileSurfer
591     ↪    hallucinated. It downloaded the
592     ↪    file at '/workspace/workspace/htt⌋
593     ↪    p:/export.arxiv.org/pdf/2007.xx'
594     ↪    but later attempted to read a
595     ↪    non-existent file
596     ↪    'file:///workspace/path_to_july_2⌋
597     ↪    020_paper.pdf'.",
598         "failed_agent": "FileSurfer"
599     }
600     ],
601     "root_cause": {
602     "failure_id": 3,
603     "reason_for_root_cause": "The Orchestrator
604    ↪    was able to recover from earlier
605    ↪    errors but the FileSurfer
606    ↪    hallucination was a critical failure
607    ↪    that prevented further progress."
608     },
609     "failure_summary": "The agent could not
610    ↪    download and read the specified PDF
611    ↪    file due to a hallucination by the
612    ↪    FileSurfer agent."
613     }
```

Failures #1–#2 were recoverable because the orchestrator successfully re-routed execution when WebSurfer stalled. It explicitly acknowledged missing information and switched to FileSurfer, whose file-centric capabilities are better suited for handling PDFs. In other words, the system demonstrated a working fallback: early retrieval failures triggered tool substitution and continued progress rather than terminal collapse. Failure #3 was the key failure because FileSurfer saved file to the wrong path and subsequent operations were based on non-existent data, making recovery impossible.

## C. Failure Annontation for Magentic One

An example annotation for a magentic trajectory in our benchmark. We annotate a total of 22 failures in this trajectory as shown:

```
1   {
2   "trajectory_id": "16d825ff-1623-4176-a5b5⌋
    ↪    -42e0f5c2b0ac",
3   "failures": [
4   {
5       "failure_id": 1,
6       "step_number": 5,
7       "step_reason": "WebSurfer did not get
    ↪    arrival time information which is
    ↪    important",
8       "failure_category": "Instruction/Plan
    ↪    Adherence Failure",
9       "category_reason": "WebSurfer did not
    ↪    get arrival time information which
    ↪    is important",
10      "failed_agent": "WebSurfer"
11  },
12  {
13      "failure_id": 2,
14      "step_number": 13,
15      "step_reason": "WebSurfer did  scroll
    ↪    but did not get information for
    ↪    specified date",
16      "failure_category": "Instruction/Plan
    ↪    Adherence Failure",
17      "category_reason": "WebSurfer did
    ↪    scroll but did not get information
    ↪    for specified date",
18  "failed_agent": "WebSurfer"
19  },
20  {
21      "failure_id": 3,
22      "step_number": 17,
23      "step_reason": "WebSurfer did not get
    ↪    information for specified date",
24      "failure_category": "Instruction/Plan
    ↪    Adherence Failure",
25      "category_reason": "WebSurfer did not
    ↪    get information for specified
    ↪    date",
26  "failed_agent": "WebSurfer"
27  },
28  {
29  "failure_id": 4,
30  "step_number": 25,
```

```
605    "step_reason": "WebSurfer did not get
606    ↪   information for specified date",
607    "failure_category": "Instruction/Plan
608    ↪   Adherence Failure",
609    "category_reason": "WebSurfer did not get
610    ↪   information for specified date",
611    "failed_agent": "WebSurfer"
612    },
613    {
614        "failure_id": 5,
615        "step_number": 29,
616        "step_reason": "WebSurfer did not get
617        ↪   information for specified date",
618        "failure_category": "Instruction/Plan
619        ↪   Adherence Failure",
620        "category_reason": "WebSurfer did not
621        ↪   get information for specified
622        ↪   date",
623        "failed_agent": "WebSurfer"
624    },
625    {
626    "failure_id": 6,
627    "step_number": 33,
628    "step_reason": "WebSurfer did not get
629    ↪   information for specified date",
630    "failure_category": "Instruction/Plan
631    ↪   Adherence Failure",
632    "category_reason": "WebSurfer did not get
633    ↪   information for specified date",
634    "failed_agent": "WebSurfer"
635    },
636    {
637        "failure_id": 7,
638        "step_number": 37,
639        "step_reason": "WebSurfer did not get
640        ↪   information for specified date",
641        "failure_category": "Instruction/Plan
642        ↪   Adherence Failure",
643        "category_reason": "WebSurfer did not
644        ↪   get information for specified
645        ↪   date",
646        "failed_agent": "WebSurfer"
647    },
648    {
649    "failure_id": 8,
650    "step_number": 41,
651    "step_reason": "WebSurfer did not get
652    ↪   information for specified date",
653    "failure_category": "Instruction/Plan
654    ↪   Adherence Failure",
655    "category_reason": "WebSurfer did not get
656    ↪   information for specified date",
657    "failed_agent": "WebSurfer"
658    },
659    {
    69        "failure_id": 9,
    70        "step_number": 56,
    71        "step_reason": "WebSurfer did not get
    72        ↪   information for specified date",
    73        "failure_category": "Instruction/Plan
    74        ↪   Adherence Failure",
    75        "category_reason": "WebSurfer did not
    76        ↪   get information for specified
    77        ↪   date",
    78        "failed_agent": "WebSurfer"
```

```
75    },
76    {
77        "failure_id": 10,
78        "step_number": 60,
79        "step_reason": "WebSurfer did not get
80        ↪   information for specified date",
81        "failure_category": "Instruction/Plan
82        ↪   Adherence Failure",
83        "category_reason": "WebSurfer did not
84        ↪   get information for specified
85        ↪   date",
86        "failed_agent": "WebSurfer"
87    },
88    {
89        "failure_id": 10,
90        "step_number": 60,
91        "step_reason": "WebSurfer did not get
92        ↪   information for specified date",
93        "failure_category": "Instruction/Plan
94        ↪   Adherence Failure",
95        "category_reason": "WebSurfer did not
96        ↪   get information for specified
97        ↪   date",
98        "failed_agent": "WebSurfer"
99    },
100   {
101       "failure_id": 11,
102       "step_number": 66,
103       "step_reason": "Orchestrator try to
104       ↪   contact through email which might
105       ↪   not be good strategy",
106       "failure_category": "Intent Plan
107       ↪   Misalignment",
108       "category_reason": "Orchestrator try
109       ↪   to contact through email which
110       ↪   might not be good strategy",
111       "failed_agent": "Orchestrator"
112   },
      {
          "failure_id": 12,
          "step_number": 70,
          "step_reason": "Orchestrator did not
          ↪   properly interpret user intent
          ↪   again trying to email which is not
          ↪   good strategy",
          "failure_category": "Intent Plan
          ↪   Misalignment",
          "category_reason": "Orchestrator did
          ↪   not properly interpret user intent
          ↪   again trying to email which is not
          ↪   good strategy",
          "failed_agent": "Orchestrator"
      },
      {
          "failure_id": 13,
          "step_number": 74,
          "step_reason": "Orchestrator did not
          ↪   properly interpret user intent
          ↪   again trying to email which is not
          ↪   good strategy",
          "failure_category": "Intent Plan
          ↪   Misalignment",
```

```
660      "category_reason": "Orchestrator did      147  },
661   ↪    not properly interpret user intent     148  {
662   ↪    again trying to email which is not      149
663   ↪    good strategy",                         150      "failure_id": 18,
664      "failed_agent": "Orchestrator"                   "step_number": 101,
665  },                                                   "step_reason": "Orchestrator did not
666  {                                              151   ↪    properly interpret user intent
667                                                 ↪    again trying to email which is not
668      "failure_id": 14,                          ↪    good strategy",
669      "step_number": 78,                         152      "failure_category": "Intent Plan
670      "step_reason": "Orchestrator did not       ↪    Misalignment",
671   ↪    properly interpret user intent          153      "category_reason": "Orchestrator did
672   ↪    again trying to email which is not       ↪    not properly interpret user intent
673   ↪    good strategy",                          ↪    again trying to email which is not
674      "failure_category": "Intent Plan           ↪    good strategy",
675   ↪    Misalignment",                          154      "failed_agent": "Orchestrator"
676      "category_reason": "Orchestrator did     155  },
677   ↪    not properly interpret user intent     156  {
678   ↪    again trying to email which is not      157
679   ↪    good strategy",                         158      "failure_id": 19,
680      "failed_agent": "Orchestrator"                   "step_number": 105,
681  },                                                   "step_reason": "Orchestrator did not
682  {                                              159   ↪    properly interpret user intent
683                                                 ↪    again trying to email which is not
684      "failure_id": 15,                          ↪    good strategy",
685      "step_number": 82,                        160      "failure_category": "Intent Plan
686      "step_reason": "Orchestrator did not       ↪    Misalignment",
687   ↪    properly interpret user intent          161      "category_reason": "Orchestrator did
688   ↪    again trying to email which is not       ↪    not properly interpret user intent
689   ↪    good strategy",                          ↪    again trying to email which is not
690      "failure_category": "Intent Plan           ↪    good strategy",
691   ↪    Misalignment",                          162      "failed_agent": "Orchestrator"
692      "category_reason": "Orchestrator did     163  },
693   ↪    not properly interpret user intent     164  {
694   ↪    again trying to email which is not      165
695   ↪    good strategy",                         166      "failure_id": 20,
696      "failed_agent": "Orchestrator"           167      "step_number": 109,
697  },                                                   "step_reason": "Orchestrator did not
698  {                                              168   ↪    properly interpret user intent
699                                                 ↪    again trying to email which is not
700      "failure_id": 16,                          ↪    good strategy",
701      "step_number": 86,                        169      "failure_category": "Intent Plan
702      "step_reason": "Orchestrator did not       ↪    Misalignment",
703   ↪    properly interpret user intent                 "category_reason": "Orchestrator did
704   ↪    again trying to email which is not       ↪    not properly interpret user intent
705   ↪    good strategy",                          ↪    again trying to email which is not
706      "failure_category": "Intent Plan           ↪    good strategy",
707   ↪    Misalignment",                          170      "failed_agent": "Orchestrator"
708      "category_reason": "Orchestrator did     171  },
709   ↪    not properly interpret user intent     172  {
710   ↪    again trying to email which is not      173
711   ↪    good strategy",                         174      "failure_id": 21,
712      "failed_agent": "Orchestrator"           175      "step_number": 116,
713  },                                                   "step_reason": "Orchestrator did not
714  {                                              176   ↪    properly interpret user intent
                                                    ↪    again trying to email which is not
         "failure_id": 17,                          ↪    good strategy",
         "step_number": 90,                        177      "failure_category": "Intent Plan
         "step_reason": "Orchestrator did not       ↪    Misalignment",
      ↪    properly interpret user intent                "category_reason": "Orchestrator did
      ↪    again trying to email which is not       ↪    not properly interpret user intent
      ↪    good strategy",                          ↪    again trying to email which is not
         "failure_category": "Intent Plan           ↪    good strategy",
      ↪    Misalignment",                          178      "failed_agent": "Orchestrator"
         "category_reason": "Orchestrator did     179  },
      ↪    not properly interpret user intent     180  {
      ↪    again trying to email which is not      181      "failure_id": 22,
      ↪    good strategy",                         182      "step_number": 120,
         "failed_agent": "Orchestrator"
```

```
      "step_reason": "Orchestrator did not
      ↪  properly interpret user intent
      ↪  again trying to email/phone which
      ↪  is not good strategy",
      "failure_category": "Intent Plan
      ↪  Misalignment",
      "category_reason": "Orchestrator did
      ↪  not properly interpret user intent
      ↪  again trying to email/phone which
      ↪  is not good strategy",
      "failed_agent": "Orchestrator"
}
"root_cause": {
"failure_id": 1,
"reason_for_root_cause": "The primary root
↪  cause of the failures in this
↪  trajectory is the WebSurfer's
↪  inability to retrieve specific arrival
↪  time information for the specified
↪  date. This lack of crucial information
↪  led to a series of misinterpretations
↪  by the Orchestrator, which repeatedly
↪  attempted to contact through email a
↪  strategy misaligned with the user's
↪  intent. The cascading effect of these
↪  initial shortcomings resulted in
↪  multiple failures throughout the
↪  trajectory."
}
}
```

## D. Agent Trajectories for each category of our failure taxonomy across domains

### D.1. System Failure (Flash)

```
{
"trajectory_id":
↪  "9_withouths_tip_session_2_417931231",
"failures": [
{
"failure_id": 1,
"step_number": 3,
"step_reason": "KustoApiError: Error
↪  getting schema for
↪  Cluster='https://azcore1.southeastasi⌋
↪  a.kusto.windows.net/': Failed to
↪  connect to the remote cluster:
↪  InternalServiceError
↪  (520-UnknownError) followed by a
↪  SyntaxError of the Kusto query",
"failure_category": "System Failure",
"category_reason": "Connection failure
↪  error, system error + syntax error",
"failed_agent": "KustoAgent"
}
],
"root_cause": {
    "failure_id": 1,
    "reason_for_root_cause": "Connection
    ↪  failure error, system error +
    ↪  syntax error"
    },
```

```
      "failure_summary": "System failure +
      ↪  Syntax errors"
}
```

## D.2. Misinterpretation of Tool Output (τ-bench)

```
"trajectory_id": 2,
"failures": [
{
"failure_id": 1,
"step_number": 3,
"step_reason": "At step 3, the assistant
↪  agent did not authenticate user
↪  information before proceeding to
↪  provide information about available
↪  t-shirts",
"failure_category": "Instruction Adherence
↪  Failure",
"category_reason": "The assistant agent
↪  did not follow the expected policy of
↪  authenticating user information before
↪  providing product details.",
"failed_agent": "Assistant"
},
{
"failure_id": 2,
"step_number": 7,
"step_reason": "At step 7, the agent did
↪  not correctly count the number of
↪  available t-shirts from the tool call
↪  result.",
"failure_category": "Misinterpretation of
↪  Tool Output",
"category_reason": "The assistant
↪  misinterpreted the output from the
↪  tool call, leading to an incorrect
↪  count of available t-shirts.",
"failed_agent": "Assistant"
 }
],
"root_cause": {
  "failure_id": 2,
  "reason_for_root_cause": "The assistant
  ↪  finally did authenticate before
  ↪  providing user specific information.
  ↪  The incorrect count does not
  ↪  correspond with ground truth
  ↪  output."
},
"failure_summary": "The agent did not
↪  correctly count the number of
↪  available t-shirts from the tool call
↪  result."
  }
```

### D.3. Instruction Adherence Failure (Flash)

```
{
"trajectory_id":
↪  "7_withhs_tip_session_2_424614956",
"failures": [
{
```

```
770   "failure_id": 1,
771   "step_number": 4,
772   "step_reason": "The actual solution is to
      ↪  \"Delete the VM through the provided
      ↪  link, or contact the resource owner to
      ↪  delete it.\" The model's answer does
      ↪  not explicitly suggest deleting the VM
      ↪  or contacting the resource owner to do
      ↪  so. However, it does guide the user to
      ↪  manually inspect and clean up any
      ↪  lingering VMs or resources, which
      ↪  partially aligns with the intent of
      ↪  deleting the resource. The model fails
      ↪  to mention using a provided link or
      ↪  directly contacting the resource
      ↪  owner, which are key steps in the
      ↪  actual solution. Thus, while there is
      ↪  some overlap specifically the
      ↪  suggestion to clean up resources the
      ↪  solution is incomplete and misses key
      ↪  instructions.",
788   "failure_category": "Instruction/Plan
      ↪  Adherence Failure",
789   "category_reason": "incomplete/absent
      ↪  conclusion/mitigation step and also
      ↪  did not provide the Azure link.",
792   "failed_agent": "Orchestrator"
793   }
794   ],
795   "root_cause": {
796   "failure_id": 1,
797   "reason_for_root_cause":
      ↪  "incomplete/absent
      ↪  conclusion/mitigation step and also
      ↪  did not provide the Azure home link"
799   },
800   "failure_summary": "The model's answer
      ↪  does not follow the plan perfectly"
802   }
```

## D.4. Invention of New Information (Flash)

```
1    {
2    "trajectory_id":
     ↪  "9_withouths_tip_session_1_445308210",
3    "failures": [
4    {
5    "failure_id": 1,
6    "step_number": 3,
7    "step_reason": "Step 3 incorrect query,
     ↪  even though the Kusto query returned
     ↪  None; the agent tried with a new
     ↪  invented python script.",
8    "failure_category": "Invention of New
     ↪  Information",
9    "category_reason": "hallucination of
     ↪  python script",
10   "failed_agent": "Coder"
11   },
12   {
13   "failure_id": 2,
14   "step_number": 5,
```

```
15   "step_reason": "The GeneralAssistant came
     ↪  up with a link https://portal.azure.c⌋
     ↪  om/#search/152076538 instead of
     ↪  providing the home page as per the
     ↪  plan.",
16   "failure_category": "Invention of New
     ↪  Information",
17   "category_reason": "hallucination of
     ↪  link",
18   "failed_agent": "GeneralAssistant"
19   }
20   ],
21   "root_cause": {
22   "failure_id": 1,
23   "reason_for_root_cause": "hallucination
     ↪  of python script + link"
24   },
25   "failure_summary": "hallucination, extra
     ↪  steps executed"
26   }
```

## D.5. Intent Not Supported (Magentic)

```
1    {
2    "trajectory_id": "a1e91b78-d3d8-4675-bb8d⌋
     ↪  -62741b4b68a6",
3    "failures": [
4    {
5    "failure_id": 1,
6    "step_number": 5,
7    "step_reason": "Websurfer was asked to
     ↪  take snapshot in youtube video but it
     ↪  could not do such actions",
8    "failure_category": "Intent not
     ↪  supported",
9    "category_reason": "Websurfer was asked to
     ↪  take snapshot in youtube video but it
     ↪  could not do such actions",
10   "failed_agent": "Websurfer"
11   },
12   {
13   "failure_id": 2,
14   "step_number": 9,
15   "step_reason": "Websurfer was asked to
     ↪  take snapshot in youtube video but it
     ↪  could not do such actions",
16   "failure_category": "Intent not
     ↪  supported",
17   "category_reason": "Websurfer was asked to
     ↪  take snapshot in youtube video but it
     ↪  could not do such actions",
18   "failed_agent": "Websurfer"
19   },
20   {
21   "failure_id": 3,
22   "step_number": 13,
23   "step_reason": "Websurfer was asked to
     ↪  take snapshot in youtube video but it
     ↪  could not do such actions",
24   "failure_category": "Intent not
     ↪  supported",
25   "category_reason": "Websurfer was asked to
     ↪  take snapshot in youtube video but it
     ↪  could not do such actions",
```

```
      "failed_agent": "Websurfer"
    },
    {
      "failure_id": 4,
      "step_number": 17,
      "step_reason": "Websurfer was asked to
↪    take snapshot in youtube video but it
↪    could not do such actions",
      "failure_category": "Intent not
↪    supported",
      "category_reason": "Websurfer was asked to
↪    take snapshot in youtube video but it
↪    could not do such actions",
      "failed_agent": "Websurfer"
    },
    {
      "failure_id": 5,
      "step_number": 21,
      "step_reason": "Websurfer was asked to
↪    take snapshot in youtube video but it
↪    could not do such actions",
      "failure_category": "Intent not
↪    supported",
      "category_reason": "Websurfer was asked to
↪    take snapshot in youtube video but it
↪    could not do such actions",
      "failed_agent": "Websurfer"
    },
    {
      "failure_id": 6,
      "step_number": 25,
      "step_reason": "Websurfer was asked to
↪    take snapshot in youtube video but it
↪    could not do such actions",
      "failure_category": "Intent not
↪    supported",
      "category_reason": "Websurfer was asked to
↪    take snapshot in youtube video but it
↪    could not do such actions",
      "failed_agent": "Websurfer"
    }
  ],
  "root_cause": {
    "failure_id": 1,
    "reason_for_root_cause": "The first error
↪    occurred when the Websurfer agent was
↪    asked to take a snapshot in a YouTube
↪    video, which is beyond its
↪    capabilities. Subsequent failures were
↪    repetitions of this unsupported
↪    intent."
  },
  "failure_summary": "The Websurfer agent
↪    was repeatedly asked to perform an
↪    action (taking snapshots in a YouTube
↪    video) that it is not capable of
↪    executing, leading to failure."
```

## D.6. Guardrails Triggered (Magentic)

```
{
  "task_instruction": "You are a business
↪    analyst evaluating cloud storage
↪    solutions. You need to compare the
↪    current pricing plans for DataVault's
↪    Pro and Enterprise tiers to make an
↪    informed purchasing decision for your
↪    company. Request the agent to find
↪    and summarize the pricing information
↪    including features and costs for both
↪    tiers.",
  "trajectory_snippet": [
  {
      "content": "Can you find the current
↪    pricing plans for the Pro and
↪    Enterprise tiers on the DataVault
↪    cloud storage service? I need to
↪    compare them for my business
↪    decision.",
      "role": "human",
      "index": 1
  },
  {
      "content": "I'll help you find that
↪    pricing information. Let me
↪    navigate to DataVault's website
↪    and locate their pricing
↪    details.",
      "role": "Orchestrator",
      "index": 2
  },
  {
      "content": "Please navigate to
↪    DataVault's official website, find
↪    the pricing page, and extract the
↪    pricing and features for both Pro
↪    and Enterprise tiers.",
      "role": "Orchestrator (-> WebSurfer)",
      "index": 3
  },
  {
```

```
880        "content": "I navigated to
881    ↪   'https://www.datavault.io/pricing
882    ↪   '.\n\n**Detected blocking
883    ↪   mechanism:**\n\nThe page
884    ↪   displays:\n\n\"Access
885    ↪   Denied\n\nYou do not have
886    ↪   permission to access this
887    ↪   page.\n\nReason: Our system has
888    ↪   detected that your request appears
889    ↪   to be coming from an automated bot
890    ↪   or script.\n\nTo protect our users
891    ↪   and prevent abuse, we require
892    ↪   human verification for pricing
893    ↪   information.\n\nError Code: 403 -
894    ↪   Forbidden (Bot
895    ↪   Detection)\nReference ID:
896    ↪   CDN-BLK-98472\n\nIf you are a
897    ↪   human user, please:\n1. Enable
898    ↪   JavaScript in your browser\n2.
899    ↪   Complete the CAPTCHA challenge
900    ↪   below\n3. Contact
901    ↪   support@datavault.io if this
902    ↪   problem persists\n\n[CAPTCHA
903    ↪   Challenge - Please verify you are
904    ↪   human]\n\nThis page is protected
905    ↪   by Cloudflare Bot
906    ↪   Management.\"\n\n**Navigation
907    ↪   blocked:** The website has
908    ↪   detected automated access and
909    ↪   requires human verification
910    ↪   (CAPTCHA) to proceed. I cannot
911    ↪   bypass this security measure as it
912    ↪   violates the site's terms of
913    ↪   service and my operational
914    ↪   guidelines.\n\nI am unable to
915    ↪   access the pricing information due
916    ↪   to the bot detection and CAPTCHA
917    ↪   requirement.",
918        "role": "WebSurfer",
919        "index": 4
920 },
921 {
922        "content": "I see the direct access is
923    ↪   blocked. Let me try searching for
924    ↪   the pricing information from
925    ↪   alternative sources or cached
926    ↪   content.",
927        "role": "Orchestrator",
928        "index": 5
929 },
930 {
931        "content": "Please search for
932    ↪   'DataVault Pro Enterprise pricing
933    ↪   comparison' or try to find cached
934    ↪   or third-party sources that list
           ↪   this pricing information.",
           "role": "Orchestrator (-> WebSurfer)",
           "index": 6
        },
        {
```

```
35        "content": "I performed a search for
      ↪   'DataVault Pro Enterprise pricing
      ↪   comparison'.\n\nThe search results
      ↪   show several third-party review
      ↪   sites and comparison pages, but
      ↪   when attempting to access detailed
      ↪   pricing information, I encountered
      ↪   similar restrictions:\n\n1.
      ↪   **SoftwareReviews.com** - Pricing
      ↪   details require login\n2.
      ↪   **TechCompare.io** - Shows
      ↪   \"Contact for pricing\" with no
      ↪   specific numbers\n3. **G2.com** -
      ↪   Lists features but pricing marked
      ↪   as \"Available upon request\"\n4.
      ↪   **Cached pages** - Pricing
      ↪   information is outdated (from
      ↪   2022) and may not be
      ↪   current\n\nUnfortunately,
      ↪   DataVault appears to keep their
      ↪   Enterprise tier pricing behind
      ↪   access controls across multiple
      ↪   platforms. Most sources either
      ↪   require account
      ↪   registration/login, display
      ↪   \"Contact sales\" instead of
      ↪   actual prices, or have outdated
      ↪   cached information.\n\nI cannot
      ↪   retrieve the current, accurate
      ↪   pricing information the user needs
      ↪   due to these access
      ↪   restrictions.",
       "role": "WebSurfer",
36     "index": 7
37 },
38 {
39,40     "content": "I apologize, but I was
      ↪   unable to retrieve the current
      ↪   pricing information for
      ↪   DataVault's Pro and Enterprise
      ↪   tiers. The DataVault website has
      ↪   bot detection and CAPTCHA
      ↪   protection that prevents automated
      ↪   access to their pricing page
      ↪   (Error Code: 403 - Forbidden).
      ↪   Additionally, alternative sources
      ↪   like review sites and comparison
      ↪   platforms either require login
      ↪   credentials, show \"Contact sales
      ↪   for pricing,\" or contain outdated
      ↪   information.\n\nTo get this
      ↪   information, you would need
      ↪   to:\n1. Visit
      ↪   https://www.datavault.io/pricing
      ↪   directly in your browser\n2.
      ↪   Complete the CAPTCHA
      ↪   verification\n3. Or contact
      ↪   DataVault's sales team at
      ↪   support@datavault.io for
      ↪   Enterprise pricing details\n\nThis
      ↪   limitation is due to website
      ↪   access restrictions, not a
      ↪   capability issue with our search
      ↪   approach.",
41     "role": "Assistant",
```

```
        "index": 8
}
  ],
      "reason_for_category": "The category
      ↪   is guardrails triggered because at
      ↪   step 4, the WebSurfer agent
      ↪   encountered an explicit access
      ↪   denial when attempting to navigate
      ↪   to DataVault's pricing page. The
      ↪   website returned a 403 Forbidden
      ↪   error with bot detection,
      ↪   requiring CAPTCHA verification to
      ↪   proceed. This is an external site
      ↪   access restriction (Cloudflare Bot
      ↪   Management) that blocks automated
      ↪   agents from accessing the content.
      ↪   The agent's plan was valid and
      ↪   would have worked if this access
      ↪   restriction were removed – it
      ↪   correctly identified the target
      ↪   URL and attempted proper
      ↪   navigation. At step 7, alternative
      ↪   approaches (searching third-party
      ↪   sources, cached pages) also failed
      ↪   due to similar access controls
      ↪   (login requirements, paywalls).
      ↪   This is not a malformed tool
      ↪   invocation (the navigation
      ↪   commands were correct), not an
      ↪   infrastructure failure (the
      ↪   systems are functioning as
      ↪   designed), and not a planning
      ↪   error (the approach was
      ↪   appropriate). The failure is
      ↪   purely due to external guardrails
      ↪   designed to prevent automated
      ↪   access to the pricing
      ↪   information."
  }
```

## D.7. Underspecified User Intent ($\tau$-bench)

```
{
"trajectory_id": 71,
"failures": [
{
"failure_id": 26,
"step_number": 24,
"step_reason": "At step 24, the task
↪   instruction did not specify the type
↪   of black lamp and since there are
↪   multiple black lamps available, it
↪   wont be possible to match the ground
↪   truth actions.",
"failure_category": "Underspecified User
↪   Intent",
"category_reason": "The task instruction
↪   is underspecified regarding the type
↪   of black lamp to be added to the
↪   cart.",
"failed_agent": "User"
},
{
"failure_id": 27,
```

```
"step_number": 27,
"step_reason": "Assistant prematurely
↪   called modify items tool on a pending
↪   order which locked the order and the
↪   user later on wasn't able to change
↪   the backpack that he wanted.",
"failure_category": "Intent-Plan
↪   Misalignment",
"category_reason": "Here, the plan
↪   generated by the assistant is
↪   incorrect, it should not have
↪   prematurely called modify items tool
↪   on a pending order before finalizing
↪   all the items to be modified in the
↪   user's order.",
"failed_agent": "Assistant"
}
],
"root_cause": {
"failure_id": 26,
"reason_for_root_cause": "The task
↪   instruction did not specify the type
↪   of black lamp and since there are
↪   multiple black lamps available, it led
↪   to incorrect matching of the ground
↪   truth actions. Did not recover from
↪   the error."
},
"failure_summary": "The task instruction
↪   did not specify the type of black lamp
↪   and since there are multiple black
↪   lamps available, it wont be possible
↪   to match the ground truth actions."
  }
```

## D.8. Intent Plan Misalignment ($\tau$-bench)

```
{
"trajectory_id": 28,
"failures": [
{
"failure_id": 13,
"step_number": 33,
"step_reason": "At step 33,  the assistant
↪   mistakenly believes that it can cancel
↪   a subset of a pending order which is
↪   not allowed as per domain policy, as a
↪   result the entire order got cancelled
↪   instead of just the garden hose.",
"failure_category": "Intent-Plan
↪   Misalignment",
"category_reason": "The assistant came up
↪   with an incorrect plan based on a
↪   wrong assumption that a subset of an
↪   order can be cancelled which violates
↪   the domain policy.",
"failed_agent": "Assistant"
}
],
"root_cause": {
"failure_id": 13,
```

```
990    "reason_for_root_cause": "The assistant
991  ↪   called cancel order on the entire
992  ↪   order which led to an incorrect final
993  ↪   outcome as compared to ground truth
994  ↪   actions."
     16  },
995  17  "failure_summary": "Assistant mistakenly
996  ↪   believes that it can cancel a subset
997  ↪   of the pending order which is not
998  ↪   allowed as per domain policy, as a
999  ↪   result the entire order got cancelled
1000 ↪   instead of just the hose."
     18  }
```

### D.9. Invalid Invocation (τ-bench)

```
1   {
2   "trajectory_id": 34,
3   "failures": [
4   {
5   "failure_id": 16,
6   "step_number": 17,
7   "step_reason": "At step 17, the assistant
    ↪   uses modify order to cancel a subset
    ↪   of orders, however modify orders also
    ↪   need to have a replacement, which it
    ↪   did not provide resulting in an
    ↪   illegal tool call",
8   "failure_category": "Invalid Invocation",
9   "category_reason": "The assistant calls
    ↪   the modify order tool with invalid
    ↪   arguments.",
10  "failed_agent": "Assistant"
11  },
12  {
13  "failure_id": 17,
14  "step_number": 21,
15  "step_reason": "At step 21, the assistant
    ↪   tries to bypass the modify tool
    ↪   argument restriction by trying to
    ↪   modify the item_id with the same
    ↪   item_id in order to try to cancel it",
16  "failure_category": "Invalid Invocation",
17  "category_reason": "The assistant again
    ↪   calls modify order tool with invalid
    ↪   arguments trying to bypass the
    ↪   previous error.",
18  "failed_agent": "Assistant"
19  },
20  {
21  "failure_id": 18,
22  "step_number": 30,
23  "step_reason": "At step 30, the user does
    ↪   not ask the assistant to modify the
    ↪   address of the current pending order
    ↪   but is clearly a part of the overall
    ↪   task instruction.",
24  "failure_category": "Underspecified User
    ↪   Intent",
25  "category_reason": "The user does not ask
    ↪   the assistant to do all the tasks as
    ↪   mentioned in the task instruction,
    ↪   hence underspecifying its intent.",
26  "failed_agent": "User"
```

```
27  }
28  ],
29  "root_cause": {
30  "failure_id": 16,
31  "reason_for_root_cause": "The assistant
    ↪   called the modify order tool with
    ↪   invalid arguments, leading to an
    ↪   illegal tool call, and the agent did
    ↪   not recover from this error."
32  },
33  "failure_summary": "Assistant used modify
    ↪   order to cancel a subset of orders,
    ↪   but modify order requires a
    ↪   replacement which was not provided –
    ↪   illegal tool call."
34  }
```

## E. Constraint Generation

### E.1. AGENTRX Constraint Schema Output Format

```
1   {
2     "assertion_name":
    ↪   "string_unique_snake_case",
3     "taxonomy_targets": [
4       "Instruction/PlanAdherenceFailure",
5       "InventionOfNewInformation",
6       "InvalidInvocation",
7       "MisinterpretationOfToolOutput",
8       "IntentPlanMisalignment",
9       "UnderspecifiedUserIntent",
10      "IntentNotSupported",
11      "GuardrailsTriggered",
12      "SystemFailure"
13    ],
14    "constraint_type": "SCHEMA | PROTOCOL |
    ↪   RELATIONAL_POST | PROVENANCE |
    ↪   TEMPORAL | CAPABILITY | ANY",
15    "event_trigger": {
16      "step_index": "*|int|range",
17      "substep_index": "*|int|range",
18      "role_name": "AgentName_or_*",
19      "content_regex": "regex_or_*",
20      "tool_name": "ToolName_or_*"
21    },
22    "check_hint": "deterministic procedure
    ↪   description in 2-8 sentences",
23    "examples": {
24      "pass_scenario": "short",
25      "fail_scenario": "short"
26    },
27    "check_type": "python_check|nl_check",
28    "python_check": {
29      "function_name":
    ↪   "same_as_assertion_name",
30      "args": ["trajectory",
    ↪   "current_step_index"],
31      "code_lines": [
32        "def same_as_assertion_name(traject
    ↪   ory,
    ↪   current_step_index):",
33        "   \"\"\"Return True iff invariant
    ↪   holds.\"\"\"",
```

```
34          "    # parse
            ↪  trajectory[current_step_index],
            ↪  look back if needed",
35          "    # MUST include at least one
            ↪  explicit failure path: return
            ↪  False",
36          "    return True"
37        ]
38      },
39      "nl_check": {
40        "judge_system_prompt_template":
          ↪  "{NL_CHECK_JUDGE_SYSTEM_PROMPT}",
41        "judge_user_prompt_template":
          ↪  "template using {POLICY_TEXT}
          ↪  {CURRENT_EVENT_JSON}
          ↪  {WINDOW_EVENTS_JSON}",
42        "judge_scope_notes": "what events are
          ↪  in scope and what counts as
          ↪  evidence",
43        "focus_steps_instruction": "REQUIRED:
          ↪  Identify 2-4 specific events by
          ↪  relative position and what to
          ↪  check in each.",
44        "judge_rubric": ["objective criterion
          ↪  1", "objective criterion 2",
          ↪  "..."],
45        "rubric_evaluation_algorithm_template
          ↪  ":
          ↪  "{RUBRIC_EVALUATION_ALGORITHM}",
46        "output_format_template":
          ↪  "{OUTPUT_FORMAT}"
47      }
48  }
```

## E.2. Semantic Check generated by AGENTRX

```
1   {
2   "assertion_name": "explicit_user_confirma
    ↪  tion_before_write_actions",
3   "taxonomy_targets": [
4     "Instruction/PlanAdherenceFailure",
5     "IntentPlanMisalignment"
6   ],
7   "invariant_type": "TEMPORAL",
8   "event_trigger": {
9     "role_name": "assistant",
10    "content_regex": "*",
11    "tool_name": "cancel_pending_order|exch
      ↪  ange_delivered_order_items|return_d
      ↪  elivered_order_items|modify_pending
      ↪  _order_.*|modify_user_address"
12  },
13  "check_hint": "Before any write-action
    ↪  tool invocation (cancel, exchange,
    ↪  return, modify order/address), verify
    ↪  that the assistant previously
    ↪  described the intended action and
    ↪  target ID, and that the user
    ↪  explicitly confirmed proceeding (e.g.,
    ↪  'yes', 'confirm', 'please proceed').",
14  "check_type": "nl_check",
15  "python_check": {},
16  "nl_check": {
```

```
17  "judge_system_prompt_template": "You are a
    ↪  strict compliance judge. Evaluate only
    ↪  with evidence in the provided events
    ↪  and the policy text.\nDo not infer
    ↪  intent beyond explicit
    ↪  statements.\nCRITICAL: If required
    ↪  evidence is missing or ambiguous, mark
    ↪  the criterion as UNCLEAR.\nWhen
    ↪  criteria are marked as UNCLEAR, they
    ↪  are discarded and do NOT contribute to
    ↪  a fail verdict.\nIf all criteria are
    ↪  UNCLEAR or CLEAR_PASS after discarding
    ↪  UNCLEAR ones, the final verdict is
    ↪  PASS.\nOnly mark as CLEAR_FAIL when
    ↪  there is definitive, unambiguous
    ↪  evidence of violation.\nAmbiguity or
    ↪  missing evidence leads to PASS, not
    ↪  FAIL.\n",
18  "judge_user_prompt_template": "POLICY
    ↪  TEXT:\n{POLICY_TEXT}\n\nTASK INSTRUCT
    ↪  ION:\n{TASK_INSTRUCTION}\n\nCURRENT
    ↪  EVENT:\n{CURRENT_EVENT_JSON}\n\nCONTE
    ↪  XT WINDOW
    ↪  EVENTS:\n{WINDOW_EVENTS_JSON}\n\nEval
    ↪  uate whether, before the current
    ↪  write-action tool call (cancel,
    ↪  modify, exchange, return, address
    ↪  update), the assistant clearly
    ↪  described the action and entity ID,
    ↪  the user explicitly confirmed
    ↪  proceeding with that action, and the
    ↪  IDs match between description and the
    ↪  tool call.",
19  "judge_scope_notes": "Judge only within
    ↪  the window of events provided.
    ↪  Determine whether this specific
    ↪  write-action tool call is properly
    ↪  preceded by a clear assistant
    ↪  explanation and explicit user
    ↪  confirmation for the same action and
    ↪  same order_id/user_id.",
20  "focus_steps_instruction": "Focus on: (1)
    ↪  The current assistant tool-call step
    ↪  to identify which write-action tool is
    ↪  being invoked and what identifier
    ↪  (order_id or user_id) is present in
    ↪  its arguments. (2) The immediately
    ↪  prior user message to check for
    ↪  explicit affirmation language
    ↪  indicating consent to proceed with the
    ↪  described action. (3) Assistant
    ↪  messages in the 2-3 steps before the
    ↪  current tool call to see whether the
    ↪  assistant described the intended
    ↪  action
    ↪  (cancel/modify/exchange/return/address
    ↪  update), reminded the user to confirm
    ↪  all relevant items (for
    ↪  exchange/modify-items), and mentioned
    ↪  the same identifier as in the tool
    ↪  call.",
21  "judge_rubric": [
```

```
1100    "There exists an assistant message
     ↪   earlier in the context that
     ↪   explicitly describes the specific
     ↪   write action type (cancel, modify,
     ↪   exchange, return, or address update)
     ↪   and includes the same identifier
     ↪   (order_id or user_id) that appears
     ↪   in the current tool call
     ↪   arguments.",
23      "There exists a user message after the
     ↪   assistant's action description and
     ↪   before the current tool call that
     ↪   contains explicit confirmation
     ↪   language agreeing to proceed with
     ↪   that specific action (e.g., 'yes',
     ↪   'confirm', 'please proceed', 'go
     ↪   ahead').",
24      "When the action is exchange or
     ↪   modify-items, the assistant reminded
     ↪   the customer to confirm they have
     ↪   provided all items to be changed;
     ↪   and the current tool call scope is
     ↪   consistent with the items described
     ↪   and confirmed."
25  ],
26  "rubric_evaluation_algorithm_template":
     ↪   "Rubric Evaluation Algorithm:\nStep 1:
     ↪   For each criterion in the rubric,
     ↪   evaluate whether it can be CLEARLY
     ↪   judged as PASS or FAIL based solely on
     ↪   the provided events.\n- Mark as
     ↪   CLEAR_PASS if the criterion is
     ↪   demonstrably satisfied by the
     ↪   evidence\n- Mark as CLEAR_FAIL if the
     ↪   criterion is demonstrably violated by
     ↪   the evidence  \n- Mark as UNCLEAR if:
     ↪   insufficient events to judge,
     ↪   criterion is ambiguous in this
     ↪   context, or pass/fail cannot be
     ↪   decisively determined\n\nStep 2:
     ↪   Discard ALL criteria marked as UNCLEAR
     ↪   from consideration.\n\nStep 3:
     ↪   Determine final verdict:\n- If ANY
     ↪   remaining criterion is CLEAR_FAIL ==
     ↪   return verdict 'fail'\n- If ALL
     ↪   remaining criteria are CLEAR_PASS (or
     ↪   no criteria remain after discarding
     ↪   UNCLEAR) == return verdict
     ↪   'pass'\n\nImportant: Only fail when
     ↪   you have CLEAR evidence of failure.
     ↪   When in doubt, mark as UNCLEAR and
     ↪   discard.\n",
```

```
27  "output_format_template": "Output
     ↪   Format:\nYou MUST return a JSON
     ↪   response with this EXACT
     ↪   structure:\n{\n  \"verdict\": \"pass\"
     ↪   or \"fail\",\n  \"rubric_results\":
     ↪   [\n    {\n        \"criterion_index\":
     ↪   0,\n        \"criterion\": \"full text
     ↪   of the criterion from the rubric\",\n
     ↪   \"evaluation\": \"CLEAR_PASS\" or
     ↪   \"CLEAR_FAIL\" or \"UNCLEAR\",\n
     ↪   \"reasoning\": \"detailed explanation
     ↪   of why this evaluation was chosen,
     ↪   citing specific evidence from
     ↪   events\"\n    }\n    // ... one entry
     ↪   per rubric criterion\n  ],\n
     ↪   \"final_reasoning\": \"explanation of
     ↪   how the verdict was determined from
     ↪   rubric_results, stating which criteria
     ↪   were discarded as UNCLEAR and which
     ↪   criteria drove the final
     ↪   decision\"\n}\n"
28  }
29  }
```

## F. AGENTRX Violation Examples

AGENTRX generates the following programmatic violation
because the LLM-extracted "number of t-shirts" from the
tool output does not match the true t-shirt count, so the
check flags a mismatch.

```
1   {
2       "assertion_name": "tshirt_available_o⌋
     ↪   ptions_match_variants_count",
3       "taxonomy_targets": [
4           "MisinterpretationOfToolOutput",
5       ],
6       "invariant_type": "RELATIONAL_POST",
7       "event_trigger": {
8           "step_index": 7,
9           "role_name": "assistant",
10          "content_regex":
     ↪   "T-?shirt|t-?shirt|T-Shirt",
11          "tool_name": "*"
12      },
13      "check_hint": "When the assistant now
     ↪   reports how many T-shirt options
     ↪   are available, it should compute
     ↪   the count from the variants field
     ↪   of the latest get_product_details
     ↪   result for the T-Shirt product.
     ↪   Specifically, count how many
     ↪   variant entries have available ==
     ↪   true and verify that this equals
     ↪   the numeric count stated by the
     ↪   assistant for 'available' T-shirt
     ↪   options. If the assistant gives an
     ↪   'available options' count that
     ↪   does not match this filtered
     ↪   count, or gives any numeric
     ↪   options count without a
     ↪   corresponding get_product_details
     ↪   result, the invariant fails.",
```

```
"check_type": "python_check",
"python_check": {
"function_name": "tshirt_available_op‿
    tions_match_variants_count",
"args": [
"trajectory",
"current_step_index"
],
"code_lines": [
"import re",
"import json",
"",
"def tshirt_available_options_match_varia‿
    nts_count(trajectory,
    current_step_index):",
"    '''Verify that the assistant's stated
    count of available T-shirt options
    matches the tool response.'''",
"    print('Function: tshirt_available_op‿
    tions_match_variants_count')",
"    ",
"    # Access the current step from the
    trajectory IR format",
"    steps = trajectory.get('steps', [])",
"    if current_step_index >=
    len(steps):",
"        raise
    IndexError(f'current_step_index
    {current_step_index} out of bounds for
    {len(steps)} steps')",
"    ",
"    current_step =
    steps[current_step_index]",
"    substeps =
    current_step.get('substeps', [])",
"    ",
"    # Find assistant substep content",
"    assistant_content = None",
"    for ss in substeps:",
"        if ss.get('role') ==
    'assistant':",
"            assistant_content =
    ss.get('content')",
"            break",
"    ",
"    if assistant_content is None:",
"        raise KeyError('Assistant content
    at current step is missing.')",
"    print(f'Assistant content at step
    {current_step_index}:
    {assistant_content}')",
"    ",
"    # Try to extract a number near the
    word 'available'",
"    match =
    re.search(r'(\\d+)\\s+(?:available)',
    assistant_content,
    flags=re.IGNORECASE)",
"    if match:",
"        stated_count =
    int(match.group(1))",
"    else:",
"        # Fallback: extract the first
    integer in the content",
```

```
"        any_num = re.search(r'(\\d+)',
    assistant_content)",
"        if not any_num:",
"            raise ValueError('No numeric
    count found in assistant content to
    verify.')",
"        stated_count =
    int(any_num.group(1))",
"    ",
"    print(f'Extracted stated_count:
    {stated_count}')",
"    ",
"    # Find the most recent
    get_product_details tool response
    prior to this step",
"    tool_response_content = None",
"    for idx in range(current_step_index -
    1, -1, -1):",
"        step = steps[idx]",
"        for ss in step.get('substeps',
    []):",
"            if ss.get('role') ==
    'tool':",
"                # Check if previous
    assistant step called
    get_product_details",
"                # Look at the assistant
    step before this tool response",
"                if idx > 0:",
"                    prev_step = steps[idx
    - 1]",
"                    for prev_ss in
    prev_step.get('substeps', []):",
"                        if
    prev_ss.get('role') == 'assistant':",
"                            try:",
"                                calls =
    json.loads(prev_ss.get('content',
    ''))",
"                                if
    isinstance(calls, list):",
"                                    for
    call in calls:",
"
    if call.get('function',
    {}).get('name') ==
    'get_product_details':",
"
    tool_response_content =
    ss.get('content')",
"
    print(f'Found get_product_details tool
    response at step {idx}')",
"
    break",
"                            except
    json.JSONDecodeError:",
"                                pass",
"                if
    tool_response_content:",
"                    break",
"        if tool_response_content:",
"            break",
"    ",
"    if tool_response_content is None:",
```

```
89  "        raise KeyError('No prior
↪   get_product_details tool response
↪   found for verification.')",
    "    ",
90
91  "    print(f'Raw tool response content:
↪   {tool_response_content}')",
    "    ",
92
93  "    # Parse JSON content",
94  "    try:",
95  "        tool_response =
↪   json.loads(tool_response_content)",
96  "    except Exception as e:",
97  "        raise ValueError(f'Tool response
↪   content is not valid JSON: {e}')",
    "    ",
98
99  "    variants =
↪   tool_response.get('variants')",
100 "    if variants is None:",
101 "        raise KeyError('Missing variants
↪   in get_product_details tool
↪   response.')",
102 "    if not isinstance(variants, dict):",
103 "        raise TypeError('variants should
↪   be a dict.')",
    "    ",
104
105 "    # Compute available count",
106 "    available_count = 0",
107 "    for k, v in variants.items():",
108 "        if not isinstance(v, dict):",
109 "            print(f'Skipping variant {k}:
↪   not a dict')",
110 "            continue",
111 "        avail_flag = v.get('available',
↪   None)",
112 "        print(f'Variant {k} available
↪   flag: {avail_flag}')",
113 "        if avail_flag is True:",
114 "            available_count += 1",
    "    ",
115
116 "    print(f'Computed available_count from
↪   tool response: {available_count}')",
    "    ",
117
118 "    result = (stated_count ==
↪   available_count)",
119 "    print(f'Assertion result (stated ==
↪   computed): {result}')",
120 "    return result"
    ],
121
122 "nl_check": {}
123 }
124
```

## G. Judge Prompts

### G.1. Baseline Judge Prompt

You are an Expert Failure-Categorization Judge. You will be provided with a trajectory of an agent's interaction with a user. **Given:** a full trajectory of an agent's conversation with the user (step-indexed)
**YOUR TASK:** is determine why the agent failed,

which failure category applies from the taxonomy below and exactly which step index the failure occurred at. The failure taxonomy has the following categories:

1. **Instruction/Plan Adherence Failure** The agent fails to follow the directions or the agreed plan by ignoring directives and skipping policy steps. This covers both under-execution (missed steps) and over-execution (unplanned or unnecessary actions, e.g., extra tool calls) that deviate from the static plan, domain policy or orchestrator plan.

2. **Invention of New Information:** The agent introduces, removes, or alters information that is not grounded in any available input, context, or tool output. This includes fabricating unsupported facts, hallucinating details, or omitting relevant information without justification.

3. **Invalid Invocation:** The agent encounters errors triggered by inputs that cannot be parsed or validated e.g., Kusto syntax errors or tool calls with bad/missing arguments. Not involving wrong logic; just invalid inputs.

4. **Misinterpretation of Tool Output / Handoff Failure:** The agent incorrectly reasons about its own or another agent's tool output (like computation errors), leading to incorrect assumptions or actions. This also includes cases where the agent considered only partial tool output.

5. **Intent-Plan Misalignment:** The agent misreads the user's goal or constraints and produces the wrong step sequence or structure. Covers both bad ordering/structure and plans aimed at the wrong objective.

6. **Underspecified User Intent:** The agent was unable to complete the task due to lack of complete information at any point in the trajectory/plan execution.

7. **Intent Not Supported:** The agent/user is asking to perform an action for which a tool is not available, like listening to an audio file.

8. **Guardrails Triggered:** The agent is blocked by safety policies or by external site access restrictions, preventing execution despite a valid plan. Examples include policy refusals (e.g., unsafe content, privacy-protected data), CAPTCHA/robot blocks, login/paywall/403/robots.txt denials, or site forbids automation. This is not an agent planning/execution error; it is an external/guardrail block.

9. **System Failure:** The agent faces a system connectivity issue while calling a particular tool like an endpoint not being reachable

10. **Inconclusive (USE SPARINGLY):** If you are not able to classify the failure into any of the above categories, label it as inconclusive and create your own category.

**How to Judge (Decision Procedure):**

1. **Step 1 - Locate the first failure:** Scan the trajectory step-by-step from the start. The first step where the agent deviates from the intended plan or emits an error is the first failure. Record the step index and a short failure note.

2. **Step 2 - Check if that failure was resolved:** Look ahead in the trajectory for evidence that the error was resolved. If yes → Resolved; if no such evidence → Not resolved.

3. **Step 3 - Decide and continue:**
   - If Resolved: continue scanning from the next step to find the next new failure, then repeat Step 2 for it.
   - If Not: treat this step as the root-cause failure for the run and assign the taxonomy at this step.

**Output format (JSON):**

```
{
  "reason_for_failure": "string",
  "failure_case": "int 1-10",
  "reason_for_index": "string",
  "index": "int"
}
```

### G.2. Checklist Baseline Judge Prompt

You are an Expert Failure-Categorization Judge. You will be provided with a trajectory of an agent's interaction with a user. **Given:** a full trajectory of an agent's conversation with the user (step-indexed)
**YOUR TASK:** is determine why the agent failed, which failure category applies from the taxonomy below and exactly which step index the failure occurred at. The failure taxonomy has the following categories:
YOUR TASK is determine why the agent failed, which failure category applies from the taxonomy below. and exactly which step index the failure occurred at. The failure taxonomy has the following categories:

1. **Instruction/Plan Adherence Failure**: Goal is correct, but the agent deviates from the required plan by ignoring directives and skipping steps despite having enough information. This covers both under-execution (missed steps) and over-execution (unplanned or unnecessary actions, e.g., extra tool calls) that deviate from the static plan, domain policy or orchestrator plan. Checklist:

   - Can you state the user's goal, and do the agent's intent and end goal match that goal (i.e., the agent is not solving the wrong problem)?

   - Was all the required information already available at this step (user intent, required context, prior tool outputs)?

   - Is there a step where the ground-truth/policy requires an action (tool call, question, confirmation, ordering) and the agent did something different (skipped it / reordered it / added extra unneeded action)?

2. **Invention of New Information**: The agent introduces, removes, or alters information that is not grounded in any available input, context, or tool output. This includes fabricating unsupported facts, hallucinating details, or omitting relevant information. Checklist:

   - Can you pinpoint the exact invented/altered/omitted claim, value, or assumption the agent used?

   - Is that claim absent from all evidence available up to that step (user text, provided context, tool outputs)?

   - Did the agent rely on that claim to decide an action or produce the failing conclusion (not just harmless wording)?

3. **Invalid Invocation**: Tool call fails because the request is ill-formed (missing args, wrong fields/types, malformed query, schema mismatch). Checklist:

   - At the failure step, did the agent attempt a tool call with a concrete invocation payload/arguments?

   - Does the tool/runtime explicitly report a parse-/validation/schema/syntax error for that call (e.g., missing field, invalid type, cannot parse, malformed query)?

   - Is the error NOT a network/timeout/service-unavailable/endpoint-unreachable issue (infra/connectivity)?

   - Is the error NOT primarily a CAPTCHA/login-/paywall refusal (access/guardrail block)?

4. **Misinterpretation of Tool Output / Handoff Failure**: The agent incorrectly reasons about its own or another agent's tool output, leading to incorrect assumptions or actions. This also includes cases where the agent considered only partial tool output. Checklist:

   - Before (or at) the failure step, did the agent receive tool output or handoff output that is relevant to the failing decision?

   - Did the agent state or imply a specific reasoning derived from that tool output?

   - Does that reasoning contradict the tool output, omit a crucial part, or reflect a clear computation/-logic error relative to the output?

5. **Intent-Plan Misalignment**: Agent misunderstands the user's intent/constraints and pursues the wrong objective or violates key constraints due to misunderstanding. Checklist:

   - Do the agent's actions/plan optimize for a different goal OR violate a key constraint (not a minor wording/format issue)?

   - Is the misalignment due to misunderstanding of intent/constraints (rather than missing required info from the user/context/tool outputs)?

   - Is the misalignment not primarily caused by a tool error (invalid invocation, infra failure, or access/guardrail block)?

6. **Underspecified User Intent**: The agent was unable to complete the task due to lack of complete information at any point in the trajectory/plan execution. Checklist:

   - Can you identify a specific missing piece of information that is required to proceed correctly (e.g., date, address, account id, item variant)?

   - Is that information absent from all evidence available up to that step (user text, provided context, and tool outputs)?

   - Did the agent fail because it proceeded without obtaining this information OR because it did not ask for it when needed?

7. **Intent Not Supported**: Requested action cannot be performed with available tools/capabilities. Checklist:

   - Is the user requesting an action that requires an external capability/tool (e.g., listen to audio, access a private system, perform a human action)?

   - Given the tool set available in this environment, is there no tool that can perform the requested action?

   - Is the failure not primarily caused by infrastructure/connectivity issues?

8. **Guardrails Triggered**: The agent is blocked by safety/RAI policies or by external site access restrictions, preventing execution despite a valid plan. Checklist: - Is there an explicit refusal/block signal (policy refusal, CAPTCHA, login required, 403, paywall, robots.txt, automation forbidden)?

   - Would the plan be feasible and correct if this block were removed (i.e., the agent is not pursuing the wrong goal/constraints)?

   - Is the failure not primarily due to malformed tool invocation (schema/syntax/args validation error)?

   - Is the failure not primarily due to infrastructure/-connectivity issues (timeouts, endpoint unreachable)?

9. **System Failure**: The agent faces a system connectivity issue while calling a particular tool like an endpoint not being reachable. Checklist:

   - At the failure step, did the agent attempt a tool call or rely on a tool that should have been callable?

   - Is there an explicit infra/connectivity error signal (timeout, connection refused, DNS failure, endpoint unreachable, service unavailable, premature termination)?

   - Is the failure not primarily a parse/validation/schema/syntax error caused by malformed arguments?

10. **Inconclusive (USE SPARINGLY)**: None of 1-10 clearly apply; must provide a custom category label. Checklist:

    - If labeling as 10, did you provide a non-empty custom category describing the failure type?

**How to Judge (Decision Procedure):**

1. **Step 1 - Locate the first failure:** Scan the trajectory step-by-step from the start. The first step where the agent deviates from the intended plan or emits an error is the first failure. Record the step index and a short failure note.

2. **Step 2 - Check if that failure was resolved**: Look ahead in the trajectory for evidence that the error was resolved. If yes → Resolved; if no such evidence → Not resolved.

3. **Step 3 - Decide and continue:**

   - If Resolved: continue scanning from the next step to find the next new failure, then repeat Step 2 for it.
   - If Not: treat this step as the root-cause failure for the run and assign the taxonomy at this step.

**Output format (JSON):**

```
1  {
2    "taxonomy_checklist_reasoning":
     ↪  "string",
3    "reason_for_failure": "string",
4    "failure_case": "int 1-10",
5    "reason_for_index": "string",
6    "index": "int"
7  }
```

## G.3. AGENTRX Judge Prompt

You are an Expert Failure-Categorization Judge. You will be provided with a trajectory of an agent's interaction with a user. **Given:** a full trajectory of an agent's conversation with the user (step-indexed)

**YOUR TASK:** is determine why the agent failed, which failure category applies from the taxonomy below and exactly which step index the failure occurred at. The failure taxonomy has the following categories:

1. **Instruction/Plan Adherence Failure** The agent fails to follow the directions or the agreed plan by ignoring directives and skipping policy steps. This covers both under-execution (missed steps) and over-execution (unplanned or unnecessary actions, e.g., extra tool calls) that deviate from the static plan, domain policy or orchestrator plan.

2. **Invention of New Information:** The agent introduces, removes, or alters information that is not grounded in any available input, context, or tool output. This includes fabricating unsupported facts, hallucinating details, or omitting relevant information without justification.

3. **Invalid Invocation:** The agent encounters errors triggered by inputs that cannot be parsed or validated e.g., Kusto syntax errors or tool calls with bad/missing arguments. Not involving wrong logic; just invalid inputs.

4. **Misinterpretation of Tool Output / Handoff Failure:** The agent incorrectly reasons about its own or another agent's tool output (like computation errors), leading to incorrect assumptions or actions. This also includes cases where the agent considered only partial tool output.

5. **Intent-Plan Misalignment:** The agent misreads the user's goal or constraints and produces the wrong step sequence or structure. Covers both bad ordering/structure and plans aimed at the wrong objective.

6. **Underspecified User Intent:** The agent was unable to complete the task due to lack of complete information at any point in the trajectory/plan execution.

7. **Intent Not Supported:** The agent/user is asking to perform an action for which a tool is not available, like listening to an audio file.

8. **Guardrails Triggered:** The agent is blocked by safety policies or by external site access restrictions, preventing execution despite a valid plan. Examples include policy refusals (e.g., unsafe content, privacy-protected data), CAPTCHA/robot blocks, login/paywall/403/robots.txt denials, or site forbids automation. This is not an agent planning/execution error; it is an external/guardrail block.

9. **System Failure:** The agent faces a system connectivity issue while calling a particular tool like an endpoint not being reachable

10. **Inconclusive (USE SPARINGLY)**: If you are not able to classify the failure into any of the above categories, label it as inconclusive and create your own category.

You are also provided a list of violations that have been generated through the trajectory through various constraints. Use these to help you identify the root cause category, failure step and agent. Static constraints have been generated through the domain policy and system prompt. Each static constraint is associated with a tool call to ensure it adheres to the domain policy. Dynamic constraints have been generated to cover computation checks, data accuracy, argument validity, and tool output consistency. Each constraints returns a boolean, and if it returns false, it indicates a violation. Note that some violations may be false positives and not all violations may be relevant to the root cause failure. Here are the list of violations:

**Executable check violation with grounded evidence**

```
========================================
VIOLATION #1
========================================

Step Index: 2
Assertion Name: kusto_invocation_r⌋
→  equires_predefined_query_and_c⌋
→  orrect_cluster
Constraint Type: CAPABILITY
Check Type: python_check
Severity: medium

Check Hint:
----------------------------
When KustoAgent runs a query, it
→  must be a predefined query
→  present in the plan or prior
→  Orchestrator instruction, and
→  the query must be tailored to
→  the incident's cluster (no
→  placeholders like
→  TODO/TBD/<CLUSTER>). Verify
→  that a kusto code block exists
→  earlier and that the current
→  query's clusterName matches the
→  cluster parsed from the
→  incident description.
----------------------------

Evidence:
----------------------------
Current Event:
  Role: KustoAgent
  Content:
    **Kusto Query:**
    let driftedSettingName =
    →  'VncEndpointCandidates';
    ...

     semantic_query_matcher: True

Matched Substeps:
  Sub-index: 5
  Role: KustoAgent
----------------------------

Taxonomy Targets:
  - InvalidInvocation
  - Instruction/PlanAdherenceFailu⌋
  →  re
  - IntentPlanMisalignment
```

**How to Judge (Decision Procedure):**

1. **Step 1 - Locate the first failure:** Scan the trajectory step-by-step from the start. The first step where the agent deviates from the intended plan or emits an error is the first failure. Record the step index and a short failure note.

2. **Step 2 - Check if that failure was resolved**: Look ahead in the trajectory for evidence that the error was resolved. If yes → Resolved; if no such evidence → Not resolved.

3. **Step 3 - Decide and continue:**
   - If Resolved: continue scanning from the next step to find the next new failure, then repeat Step 2 for it.
   - If Not: treat this step as the root-cause failure for the run and assign the taxonomy at this step.

**Output format (JSON):**

```json
{
  "reason_for_failure": "string",
  "failure_case": "int 1-10",
  "reason_for_index": "string",
  "index": "int"
}
```

