# OpenReview forum: "AgentRx: Diagnosing AI Agent Failures from Execution Trajectories"
_ICML.cc/2026/Conference — Submitted to ICML 2026_

### Official Review · Reviewer_AdLZ · 2026-03-12

**Soundness:** 2
**Presentation:** 3
**Significance:** 2
**Originality:** 2
**Overall Recommendation:** 3
**Confidence:** 4

**Summary:**

This paper studies the problem of diagnosing failures in execution trajectories of LLM-based agents. The authors introduce AGENTRX, a framework that synthesizes constraints from tool schemas, domain policies, and trajectory prefixes, evaluates these constraints during execution, and uses the resulting violation logs together with an LLM-based judge to localize the first unrecoverable failure step and assign a failure category. The paper also presents a benchmark of 115 annotated failed trajectories across three domains and a cross-domain failure taxonomy. Experiments evaluate the proposed framework on step localization and failure attribution tasks.

**Compliance With Llm Reviewing Policy:**

Affirmed.

**Final Justification:**

While the rebuttal helps narrow the discussion, concerns regarding the relatively small evaluation scale and limited subsets are not fully resolved. I therefore maintain my score.

**Key Questions For Authors:**

- The proposed benchmark is central to the paper’s contribution. Could the authors report the inter-annotator agreement statistics (e.g., Cohen’s κ or similar) for the failure step and category annotations, and describe the annotation validation procedure in more detail?

- The proposed benchmark is central to the paper’s contribution. Could the authors report the inter-annotator agreement statistics (e.g., Cohen’s κ or similar) for the failure step and category annotations, and describe the annotation validation procedure in more detail?

- Some failure categories in the taxonomy (e.g., Plan Adherence Failure vs. Intent–Plan Misalignment) appear conceptually related and may plausibly co-occur in a single step. How does the framework behave in cases where multiple failure modes contribute to a trajectory failure?

**Limitations:**

yes

**Strengths And Weaknesses:**

**Strengths**

- **Significance** The paper addresses an important and timely problem: diagnosing failures in long-horizon LLM agent trajectories, which is increasingly relevant as agent systems involve complex tool use and multi-step execution.

- **Originality** The work introduces the concept of identifying the first unrecoverable failure rather than the first observed failure in a trajectory, providing a useful perspective for analyzing agent execution errors.

- **Soundness** The proposed framework leverages constraint violations as structured diagnostic signals for downstream reasoning, which is a reasonable and interpretable approach to failure attribution in execution traces.

- **Originality** The paper combines constraint synthesis, trajectory validation, and LLM-as-a-judge reasoning into a unified diagnostic pipeline for agent execution traces.

- **Significance** The introduction of a benchmark with annotated failure steps and a cross-domain failure taxonomy provides an initial resource for studying failure localization in agent systems.

- **Presentation** The paper is generally clearly written, with a well-structured narrative and explicit definitions of failure types that make the problem setting accessible.


**Weaknesses**

- **Soundness** The empirical evaluation relies on a relatively small benchmark (115 trajectories), with some comparisons conducted on even smaller subsets (e.g., 16 trajectories for the Who&When comparison). This makes it difficult to assess the robustness and statistical reliability of the reported improvements.

- **Soundness** The benchmark is fully constructed by the authors, but important validation details are missing, such as inter-annotator agreement statistics, annotation verification procedures, and statistical significance analysis of experimental results.

- **Soundness** Although multiple ablations are reported, the analysis of experimental outcomes remains limited. For instance, differences between one-shot and step-by-step constraint generation are primarily attributed to token length, while other potentially relevant factors—such as trajectory complexity, tool usage patterns, or failure density—are not examined.

- **Soundness** The framework relies heavily on repeated LLM calls (for constraint synthesis, evaluation, and judging), yet the paper does not report inference cost, number of model calls, or token consumption. This makes it difficult to assess the computational practicality of the approach.

- **Presentation** The overall system pipeline is conceptually described but operational details remain unclear. In particular, the paper does not clearly specify how synthesized constraints are internally represented, how guards are executed, or how semantic checks interact with the LLM-based judge. A clearer architecture diagram and implementation description would improve reproducibility.

- **Presentation** Some aspects of the taxonomy implementation appear inconsistent. For example, prompts reference an “Inconclusive” category that is not reflected in the reported taxonomy statistics, which may cause confusion about how labels are handled during evaluation.

- **Originality** The proposed framework primarily combines existing components (constraint checking, rule-based validation, and LLM-as-a-judge reasoning) into a structured diagnostic pipeline. While useful in practice, the work does not introduce new learning algorithms, theoretical analysis, or formal guarantees regarding failure attribution.

- **Originality** The failure taxonomy appears to function as a fixed set of predefined categories embedded in prompts. This raises questions about how easily the taxonomy can be extended or adapted to new domains and failure types.

- **Significance** While the problem of diagnosing agent failures is important, the current evaluation scope makes it difficult to assess the broader impact of the method. Additional validation on larger or independent datasets would help clarify the generality and practical applicability of the approach.

---

> ### Author Rebuttal · Authors · 2026-03-31
>
> > The proposed benchmark is central to the paper’s contribution. Could the authors report the inter-annotator agreement statistics (e.g., Cohen’s κ or similar) for the failure step and category annotations, and describe the annotation validation procedure in more detail?
>
> To assess inter-annotator agreement during the open coding process, we conducted a validation exercise on a subset of 5 trajectories sampled from the magentic-one dataset. These trajectories were independently annotated by three annotators using the same taxonomy and annotation guidelines for both failure step identification and category labeling. Each annotator was also given examples for each of the 9 classes for reference. Since Cohen’s Kappa is defined for two annotators, we computed pairwise agreement across all annotator pairs and reported the mean. The pairwise κ scores were as follows: 1.0 (Annotator 1 vs 2), 0.706 (Annotator 1 vs 3), and 0.706 (Annotator 2 vs 3), resulting in a mean κ of 0.804. This corresponds to near perfect agreement (0.8-1.0), indicating that the taxonomy is well defined and used consistently. The only disagreement occurred in a single trajectory where one annotator labeled the category differently, suggesting a borderline case between two possibly related classes. We manually reviewed such disagreements and refined the taxonomy definitions and annotation guidelines.
>
> > Some failure categories in the taxonomy (e.g., Plan Adherence Failure vs. Intent–Plan Misalignment) appear conceptually related and may plausibly co-occur in a single step. How does the framework behave in cases where multiple failure modes contribute to a trajectory failure?
>
> Our taxonomy was derived via grounded theory coding, where categories were iteratively refined until reaching theoretical saturation. The key design principles:
>
> 1. Mutual exclusivity at the critical step: While multiple failure modes may appear across a trajectory, the critical failure step receives exactly one category label. The category captures the primary reason that specific step is results in unrecoverable failure.
>
> 2. Disambiguation criteria: Categories like Plan Adherence Failure (agent deviates from the correct plan) vs. Intent-Plan Misalignment (agent forms the wrong plan from correct intent) are distinguished by whether the plan itself is correct. We operationalize this via the taxonomy checklist K, which provides targeted yes/no questions for each category.
>
> 3. Extensibility: While the taxonomy is currently embedded as a fixed set in prompts, extending it requires only (i) defining new category questions in the checklist K and (ii) adding them to the judge prompt. The grounded theory methodology we describe can be applied to new domains to discover additional categories.
>
> The “Inconclusive” issue reflects an evaluation outcome rather than a tenth taxonomy label. The taxonomy itself has 9 categories, while the evaluation protocol allows the judge to output inconclusive (or a custom label) when the evidence does not support a reliable taxonomy assignment. We will clarify this distinction more explicitly in the revision to avoid confusion.
>
> On presentation, we respectfully think several of these details are already specified in the current draft. We agree that an implementation-oriented summary would improve reproducibility, but these operational details are not absent from the paper. Section 3 specifies the overall pipeline: the setup and normalization into a common IR, the synthesis of global and dynamic constraints (Section 3.1), the definition of a constraint as a guard plus assertion and how guards are evaluated (Section 3.2), the construction of the step-indexed validation log (Section 3.3), and how the LLM-based judge consumes the validation log together with the taxonomy checklist to predict the critical step and failure category (Section 3.4). Further, Appendix E.1 gives the concrete constraint schema output format, and Appendix E includes examples of checks.
>
> > Cost analysis:
>
> We analyzed cost under the same GPT-5 pricing assumptions used in our experiments. The baseline costs USD 0.0378 per trajectory. The best-performing version of AgentRx costs USD 0.1742 per trajectory, which is approximately 4.6× the baseline. However, this increase is concentrated almost entirely in invariant generation, which accounts for roughly 81% of the total cost, while the final judge stage remains comparable to and slightly cheaper than the baseline. In absolute terms, the cost is roughly 17 cents per trajectory.
>
> Finally, on originality and significance, we agree that the paper does not claim a new learning algorithm or formal guarantee and we do mention it anywhere. Rather, the contribution is a new benchmark for first unrecoverable critical failure attribution, a grounded cross-domain taxonomy, and an auditable diagnostic pipeline; the paper also explicitly notes that the taxonomy may require extension in new domains.

---

> > ### Author Rebuttal · Reviewer_AdLZ · 2026-04-04
> >
> > Thank you for the detailed clarification on annotation quality and taxonomy design, which helps address some of my concerns. However, the limited scale of the benchmark and small evaluation subsets still make it difficult to assess the robustness of the results, so my assessment remains unchanged.

---

> > > ### Author Response · Authors · 2026-04-05
> > >
> > > Thank you for the follow-up. We respectfully think the remaining concern is now narrower than your original review suggests: the rebuttal addressed annotation quality, clarified the taxonomy, specified the pipeline in more detail, and added cost analysis. We do not think the current scale by itself undermines the paper’s core contribution as a first benchmark, taxonomy, and auditable diagnosis framework for this problem.

---

### Official Review · Reviewer_Naeu · 2026-03-12

**Soundness:** 2
**Presentation:** 3
**Significance:** 2
**Originality:** 3
**Overall Recommendation:** 3
**Confidence:** 4

**Summary:**

This work presents a dataset of 115 annotated failed agent trajectories across 3 domains: 29 from GPT4o on \tau-bench; 42 on Flash (multi-agent incident management setting); and 44 on Magentic-One (multi-agent web and file management tasks).
Each trajectory is labeled with one critical failure step and an error category. A critical failure is defined as the first **unrecoverable** error step.
The error categories come from a taxonomy of 9 candidates that can generalize across domains and multi-agent settings.

This work also present AgentRX, a system that is able to detect such failure cases.
Given a trajectory, agent and tool json schemas, and domain-specific natural language rules, AgentRX first produces a set of constraints to satisfy (both programmatic and semantic based), then checks them step by step and produces an error log for each trajectory.
A judge LLM is then responsible for detecting the critical failed trajectory step and its error category based on the previously produced error log and a collection of yes/no validation questions for each of the 9 error categories.

Experimental results with gpt5 show how AgentRX performs and ablation studies informs what contributes to the performance.

**Compliance With Llm Reviewing Policy:**

Affirmed.

**Key Questions For Authors:**

*Reserve for questions where responses would likely change your evaluation, clarify confusing points, or address critical limitations. Number your questions.*

1. Error category 6 is _"Under-specified user intent: the agent was unable to complete the task due to lack of complete information"_. How is this an agent failure and not a user / task description issue? Same question for error category 7: _"Intent not supported"_.

2. How generalizable is your method? If it is, then you should be able to evaluate AgentRX on other's datasets such as W&W.

3. In Table 2, can you report the standard deviation across n=3 runs? With such a small test set, we may expect high variance in the results, hence lowering their significance.

4. In tables 2, 4, 5, and 6 you have a "Baseline", but is not clear at all what this baseline is? is it W&W or is it AgentRX under some specific hyperparameters?

5. Can AgentRX detect the "no-error" case? This is required to be able to run it at scale on novel un-validated trajectories.

**Limitations:**

yes.

**Strengths And Weaknesses:**

### Strengths

- The paper is well motivated, the problem is well presented and the methodology is clear. Identifying the critical step in an agent trajectory is a very important problem to study. This work proposes both a small dataset of 115 labeled trajectories

- Generating both programmatic and semantic constraints for each trajectory based on task metadata is an original idea that seems to facilitate the error detection task. Combined with a strong judge LLM, the proposed system is sound and original.

- A lot of work has been done towards both the careful dataset labeling and ablation studies on the proposed solution. In particular, the generative-based taxonomy of error codes is interesting as it does not bias the categories towards predefined human preferences. While the method is only compared to one other previous work, significant ablation studies informs the contribution of the different aspects of the AgentRX error detection system.


### Weaknesses

- It is not clear how significant the scientific contribution is. Proposing both the problem and the solution will cause implicit biases while designing the solution. To assert this is not an issue, the work should validate the solution (AgentRX) on other, independent, error labeling datasets, such as W&W. Even if the error categories will not match previous works (as they were defined specifically for this dataset), the position of the errors should still be verifiable. As such, the work would benefit from clarifying the following areas: How well does AgentRX work on other error labeling benchmarks? Is it reliable enough to automatically generate labels at scale on a wide range of domains? The AgentRX dataset is relatively small due to high human cost, but could you use AgentRX to augment it further while requiring human verification (verification is often cheaper and faster than generation). Could the dataset be used for training a better AgentRX agent?

- While the work focuses on critical step error detection, the dataset constructed should still cover successful trajectories. Any good error detection system must be able to identify the "no error" case, otherwise one can have a very strong accuracy while potentially hiding high false negative ratios.

- While the motivation and methodology is very clear and well presented, the experimental section is a bit harder to digest. For instance, referencing Table 5 early in the text is confusing as not all ablations have been explained. The word "Baseline" is used in tables 2, 4, 5, 6 but it is not clear what this refers to (see question n.4 below).

---

> ### Author Rebuttal · Authors · 2026-03-31
>
> We would like to clarify that the paper already includes both a conceptual and an experimental comparison to Who&When: Section 2 explains why our target of the first unrecoverable critical failure is more fine-grained than the first observed failure, and Section 4.2 directly evaluates against a prompt-modified W&W judge on τ-bench and Magentic-One. We will strengthen this comparison further in the revision.
>
> > 1. Error category 6 is "Under-specified user intent: the agent was unable to complete the task due to lack of complete information". How is this an agent failure and not a user / task description issue? Same question for error category 7: "Intent not supported".
>
> Under-specified User Intent is used when the trajectory becomes unrecoverable because the agent proceeds despite missing information, instead of asking the user a clarifying question or abstaining until the ambiguity is resolved. In other words, the failure is not that the user’s request was incomplete by itself, but that the agent handled that incompleteness incorrectly. Similarly, Intent Not Supported is used when the task or delegated subtask is outside the capabilities of the available tools/agents, but the system still proceeds as if it were executable instead of surfacing that limitation appropriately. We will revise these definitions to make the agent’s incorrect response to under-specification or unsupported capability more explicit.
>
> > How generalizable is your method? If it is, then you should be able to evaluate AgentRX on other's datasets such as W&W.
>
> To assess generalizability beyond our benchmark, we evaluated AgentRx on a subset of the MAST dataset. We sampled 8 traces per framework and ran the full AgentRx pipeline on traces from 6 frameworks (AG2, AppWorld, HyperAgent, Magentic-One, MetaGPT, and OpenManus). AgentRx successfully processed traces from all six frameworks without any code changes, producing both step-localization outputs and failure-category predictions. Since the taxonomies are not directly aligned, we do not claim one-to-one category comparability here; instead, this experiment evaluates whether the framework transfers structurally to a different benchmark and set of agent systems. Interestingly, AgentRx localized 53% of failures to the last third of the trajectory, which is consistent with MAST’s observation that verification and termination-related failures are a major failure class. ﻿Plan Adherence was the dominant prediction across frameworks: 6/8 for HyperAgent (SWE-bench tasks where agents ignored provided patches), 6/8 for OpenManus (research tasks where agents abandoned multi-step plans before completion), and 4/8 for MetaGPT (code generation tasks where implementations deviated from test specifications). These results suggest that AgentRx can already serve as a candidate-label generator across diverse MAS frameworks, even outside the benchmark it was introduced with. AgentRx can be used to bootstrap annotation at larger scale by first generating candidate critical steps, categories, and supporting validation evidence, and then asking humans to verify or correct them. In this setting, verification is indeed cheaper than labeling from scratch, and the verified outputs can be added back to expand the dataset over time.
>
> >  3. In Table 2, can you report the standard deviation across n=3 runs? With such a small test set, we may expect high variance in the results, hence lowering their significance.
>
> We note that the standard deviation is low (σ ≤ 3.45% for accuracy metrics and σ ≤ 3.7% for mean step error), indicating that the results are stable across 3 runs despite the test set size.
>
> > 4. In tables 2, 4, 5, and 6 you have a "Baseline", but is not clear at all what this baseline is? is it W&W or is it AgentRX under some specific hyperparameters?
>
> We will clarify that “Baseline” refers to our judge without AgentRx-added evidence, while “Baseline+Vio.” and related settings add validation signals incrementally.
>
> > 5. Can AgentRX detect the "no-error" case? This is required to be able to run it at scale on novel un-validated trajectories.
>
> The "no-error" case is effectively captured by our Inconclusive outcome: when the agent cannot identify a grounded critical failure, it can return Inconclusive rather than forcing a potentially spurious error label. We will clarify this more explicitly in the revision. We also note that one of the primary use cases of AgentRx is to assist agent developers in root-causing and localizing failures for trajectories that have already been identified as problematic, e.g., through end-user negative feedback or clear task failure. Today, this process is largely manual and requires significant human effort and domain expertise; AgentRx is designed to make this diagnosis process more systematic, auditable, and scalable.

---

> > ### Author Rebuttal · Reviewer_Naeu · 2026-04-03
> >
> > Thank you for answering all my questions and clarifying my concerns.
> > After re-evaluating I will keep my score as the method is only
> > compared to one other previous work on only 45 trajectories.
> > In addition, the proposed dataset is quite small
> > despite sugesting a method to scale it up.

---

> > > ### Author Response · Authors · 2026-04-04
> > >
> > > We are happy to compare against the full MAST dataset, but we were not able to complete that analysis within the rebuttal window. More importantly, a direct comparison to MAST is not fully apples to apples because the taxonomies differ. We therefore included the experiment mainly to show that AgentRx generalizes across domains, rather than to claim a one-to-one comparison with MAST.

---

### Official Review · Reviewer_gwQM · 2026-03-12

**Soundness:** 3
**Presentation:** 3
**Significance:** 3
**Originality:** 3
**Overall Recommendation:** 4
**Confidence:** 2

**Summary:**

The paper introduces AgentRx, which is a framework that can diagnose why agents fail automatically by localizing critical failure steps and assigning a root-cause category.

Overall, the authors:
1. Propose a benchmark of 115 failure trajectories across three domains
2. A 9-category failure taxonomy via grounded coding
3. A synthesis and LLM as a judge pipeline to produce an auditable log
4. The experiments show that their framework has a better performance compared to baselines in step localization and failure categorization.

**Compliance With Llm Reviewing Policy:**

Affirmed.

**Key Questions For Authors:**

See weaknesses.

**Limitations:**

See weaknesses.

**Strengths And Weaknesses:**

Strengths:
1. The debugging agentic system topic is important and interesting, and for sure, is underexplored. The motivation of this paper is well explained.

2. By using grounded theory coding, the framework doesn't need to manually impose a predefined taxonomy anymore, which sounds good. The pipeline starting from marking to failure identification via backward tracing is rigorous.

3. The decomposition into global and dynamic constraints is interesting.

Weaknesses
1. The number of data samples in this benchmark is limited. It raises a minor concern about scalability. A significance test will be good to further evaluate whether the improvements are from noise or advanced design.

2. The overall performance is still low( but I think it is fine). 54% means nearly half of the critical failure steps are still missing.

3. Cost analysis is needed since the overall framework relies a lot on GPT5.

4. How to distinguish the granularity, i.e. the overlap, among the taxonomy?

---

> ### Author Rebuttal · Authors · 2026-03-31
>
> > The number of data samples in this benchmark is limited. It raises a minor concern about scalability. A significance test will be good to further evaluate whether the improvements are from noise or advanced design. The overall performance is still low( but I think it is fine). 54% means nearly half of the critical failure steps are still missing.
>
> Thank you for the thoughtful review. We agree that the benchmark is still modest in size, but we did account for variance by running each configuration 3 times and reporting mean ± standard deviation. We also want to emphasize that the goal of the paper is not to claim that failure attribution is solved, but rather to show that structured validation signals substantially improve diagnosis quality over an LLM-as-a-judge baseline. For example, on τ-bench, step accuracy improves from 32.2 to 54.0 and category accuracy from 25.3 to 40.2.
>
> > Cost analysis is needed since the overall framework relies a lot on GPT5.
>
> We analyzed cost under the same GPT-5 pricing assumptions used in our experiments. The baseline costs USD 0.0378 per trajectory. The best-performing version of AgentRx costs USD 0.1742 per trajectory, which is approximately 4.6× the baseline. However, this increase is concentrated almost entirely in invariant generation, which accounts for roughly 81\% of the total cost, while the final judge stage remains comparable to and slightly cheaper than the baseline. In absolute terms, the cost is roughly 17 cents per trajectory. We would also like to point out the human annotation cost in debugging and evaluating agentic traces. AgentRx was motivated by discussions with several technology firms that are deploying AI agents at the scale of hundreds of millions of user queries per day. Improving agent quality and robustness is challenging because diagnosing failures is itself non-trivial. This is primarily due to three factors:
> 1. Effort: Agent trajectories can be long-running and contain many steps; for example, we observed an average of 33 steps in Magentic-One trajectories. Manually inspecting and annotating failures therefore requires substantial human effort. In our own experience, labeling each failed trajectory in the benchmark took about 22 minutes on average.
> 2. Scale: Even at a very low sampling rate, manually annotating thousands of trajectories per day is not feasible for agents serving millions of daily users.
> 3. Domain expertise: Annotating failures requires a thorough understanding of the system prompts, tools, domain, and user guardrails. This requires substantial investment and must continuously evolve as the agents themselves evolve over time.
>
> > How to distinguish the granularity, i.e. the overlap, among the taxonomy?
>
> Our taxonomy was derived via grounded-theory coding, with categories iteratively refined until reaching theoretical saturation. The key design principles are:
> 1. Mutual exclusivity at the critical step: While multiple failure modes may appear across a trajectory, the critical failure step receives exactly one category label. That label captures the primary reason the specific step results in an unrecoverable failure.
> 2. Disambiguation criteria: Categories such as Plan Adherence Failure (the agent deviates from the correct plan) and Intent-Plan Misalignment (the agent forms the wrong plan from the correct intent) are distinguished by whether the plan itself is correct. We operationalize this distinction through the taxonomy checklist, which provides targeted yes/no questions for each category.
> 3. Extensibility: Although the taxonomy is currently embedded as a fixed set in the prompts, extending it requires only (i) defining new category questions in the checklist K, and (ii) adding them to the judge prompt. The grounded-theory methodology we describe can likewise be applied to new domains to discover additional categories.
>
> We will strengthen the discussion of cost, include the human-cost perspective more explicitly, and clarify category boundaries and overlaps in the revision.

---

> > ### Author Rebuttal · Reviewer_gwQM · 2026-04-03
> >
> > Dear Authors,
> >
> > Thank you for your thorough and well-organized rebuttal. I appreciate the effort you put into addressing each of my concerns.
> >
> > After carefully reviewing your responses, I have decided to maintain my original scores. While I acknowledge the additional clarifications and experiments you have provided, I feel that my initial assessment still reflects my overall evaluation of the work.

---

### Official Review · Reviewer_pHwg · 2026-03-12

**Soundness:** 3
**Presentation:** 3
**Significance:** 2
**Originality:** 2
**Overall Recommendation:** 3
**Confidence:** 3

**Summary:**

This paper introduces AgentRx, a framework for diagnosing failures in AI agent execution trajectories. The authors create a benchmark of 115 failed trajectories across three domains (API workflows, incident management, and web/file tasks), each annotated with a critical unrecoverable failure step and a taxonomy label derived via grounded theory. AgentRx, automatically synthesises constraints from tool schemas, policies, and trajectory prefixes, generates violation logs, and uses an LLM-based judge to localise the root cause and categorise the failure.
Empirically, the approach improves step localisation and failure categorisation over baselines and prior work in some domains, particularly τ-bench. However, improvements vary across settings, and comparisons to closely related work (especially Who&When and other Magentic-based diagnostics) raise questions about novelty and evaluation scope.

**Compliance With Llm Reviewing Policy:**

Affirmed.

**Final Justification:**

The authors clarified and justified originality, so the scores reflect that.

**Key Questions For Authors:**

1. Why was there no direct comparison to the MAST LLM annotator, especially given the shared use of Magentic-One trajectories? How does AgentRx differ?
2. Given the small number of trajectories in several domains (e.g., 29 and 16 samples), have you conducted statistical tests to assess whether improvements are significant rather than sampling artefacts?
3. Is there any error analysis that you have done that can strengthen the conclusions? In cases where AgentRx underperforms baseline (e.g., some Magentic settings), what patterns explain this? Are violation logs sometimes misleading or overly sparse?

**Limitations:**

Yes

**Strengths And Weaknesses:**

Soundness
The work is technically sound overall. The grounded theory approach is well executed in my opinion and good to see the use of SMEs for annotations compared to many papers using just LLMs-as-a-judge. Running each configuration multiple times to measure variability is good practice and increases confidence in the reported results. The judges are clearly documented in the appendix, which supports transparency.
However, the experimental volumes are small in several key comparisons (e.g., 29 and 16 samples), making it unclear whether the improvements are statistically meaningful. The ablations do not clearly establish which components are most responsible for gains, and in some cases the results appear inconclusive.

Originality
This is the weakest part. The main idea is not convincingly differentiated from closely related work, particularly https://arxiv.org/pdf/2503.13657. It is unclear what is fundamentally new beyond that prior approach. The paper does not compare against the MAST LLM annotator, despite both using Magentic-One trajectories, which weakens claims of novelty. It is also not clear whether the notion of “critical failures” leads to a meaningful difference in practice compared to what MAST already proposes.

Significance
The paper addresses an important and practical problem: diagnosing failures in long-horizon agent trajectories. The benchmark and taxonomy could be useful for future debugging work.
That said, the small evaluation sizes limit confidence in the strength of the empirical claims, and the reported gains are modest and domain-dependent. As presented, the impact appears incremental rather than transformative.

Presentation
The paper is clearly written and generally easy to follow. The taxonomy is well described, and the appendix provides substantial detail on judges and examples.
However, the comparison to prior work is limited, particularly with respect to the most closely related approaches. The appendix includes many examples (may be too many) but relatively little synthesis or higher-level analysis of patterns in the results.

---

> ### Author Rebuttal · Authors · 2026-03-31
>
> > Why was there no direct comparison to the MAST LLM annotator, especially given the shared use of Magentic-One trajectories? How does AgentRx differ?
>
> Thank you for the thoughtful review. We would like to clarify that MAST and AgentRx are designed for different annotation targets and operate at different levels of abstraction. MAST is a system-level taxonomy of recurring multi-agent failure patterns, intended to characterize how MAS fail in aggregate and to guide system design and analysis. By contrast, AgentRx is a root-cause taxonomy for the first unrecoverable critical failure in an execution trajectory: it identifies the earliest failure from which the agent does not recover, annotates the corresponding critical step, and assigns a category explaining why the run became unrecoverable. This distinction is central to debugging and repair. This also means that the two taxonomies are not in one-to-one correspondence. MAST focuses on broader system-level patterns, whereas AgentRx isolates step-level root-cause categories needed for diagnosis. As a result, some AgentRx labels have only partial analogs in MAST, while others—such as Invalid Invocation, Intent Not Supported, Guardrails Triggered, and System Failure—do not appear as standalone MAST labels. We therefore view the two as complementary rather than interchangeable: MAST helps explain what kinds of failures recur in MAS, while AgentRx helps identify where a particular run first became unrecoverable and why. **That said, we ran the full AgentRx pipeline on a subset of the MAST dataset spanning 6 frameworks, and AgentRx successfully produced step-localization outputs and failure-category predictions without any code changes**. Since the taxonomies are not directly aligned, we do not claim one-to-one category comparability there; rather, we use that experiment to show that the framework transfers structurally across agent systems. Interestingly, AgentRx localized 53% of failures to the last third of the trajectory, which is consistent with MAST’s observation that verification and termination-related failures are a major failure class. ﻿Plan Adherence was the dominant prediction across frameworks: 6/8 for HyperAgent, 6/8 for OpenManus, and 4/8 for MetaGPT.
>
> We would also like to clarify that the paper already includes both a conceptual and an experimental comparison to prior work. In Section 2, we explain why Who&When’s labeled failure is not necessarily the first unrecoverable critical failure under our definition. In Section 4.2, we directly compare against a prompt-modified W&W judge on τ-bench and Magentic-One by adapting it to predict the first unrecoverable critical step.
>
> On the empirical side, we would like to clarify that the evaluation is not limited: the benchmark contains 115 failed trajectories across domains, which provides a reasonably substantive basis for evaluation. That said, we agree that a few of the subsets are smaller. To address variability, we ran each configuration multiple times and reported mean ± standard deviation. We also agree that the paper would be strengthened by adding statistical tests and by expanding the error analysis, especially in cases where AgentRx underperforms the baseline or where validation signals are sparse or noisy.
>
> > Error analysis:
>
> Broadly, the errors fall into two categories:
>
> 1. Surfaced violations misdirect the judge toward an incorrect error category or toward an inconclusive diagnosis.
>
> 2. Surfaced violations capture downstream symptoms or secondary failures rather than the true root cause
>
> Flash: The correct label is System Failure, since the underlying problem is infrastructural: the Kusto query failed because of a backend or endpoint issue rather than because the agent chose an invalid action. Concretely, the relevant step is better read as something like KustoAgent -> run_kusto_query(cluster=..., query=...) -> network/endpoint error, not as an invalidly formed tool call. However, the violation log presented to the judge is dominated by invocation-oriented checks, such as assertions about predefined queries, correct cluster selection, endpoint validity, and repeated retries. This case highlights a limitation of the current setup: when similar low-level assertions can be triggered by both tool misuse and backend failure, the judge may over-attribute the problem to the agent.
>
> Magentic-One (task-id=﻿1f975693-876d-457b-a649-393859e79bf): ﻿The correct label is Invention of New Information -- the Orchestrator hallucinated that a PDF was successfully downloaded and issued instructions based on that false assumption. AgentRx checklist_ focused on a late-trajectory guardrail block at step 51 ("ResponsibleAIPolicyViolation... the Orchestrator nevertheless emits FINAL ANSWER that is ungrounded"), shifting the diagnosis to Guardrails Triggered because the violation evidence pointed at a downstream symptom rather than the upstream hallucination.

---

> > ### Author Rebuttal · Reviewer_pHwg · 2026-04-02
> >
> > Thank you to the authors for their detailed and thoughtful responses in the rebuttal. We appreciate the clarification provided.
> > The authors have offered substantial additional detail explaining how their work introduces a novel concept compared to MAST. Based on this clarification, I would increase the originality score. At the same time, the authors largely acknowledged the evaluation limitations we previously highlighted, which continue to affect the overall significance of the results, so my assessment in that regard remains unchanged.

---

> > > ### Author Response · Authors · 2026-04-02
> > >
> > > Apologies for the confusion, we do not mean to acknowledge the limitation, we would like to clarify that the evaluation is not limited: the benchmark contains 115 failed trajectories across domains, which provides a reasonably substantive basis for evaluation. We are also happy to add results with the MAST comparison in the main paper. The previous related works have augmented their dataset with LLM annotators whereas our dataset is annotated entirely by humans so even a dataset of size 115 has substantial significance (in addition we do provide variability guarantees as well).

---

### Decision · Program_Chairs · 2026-04-30

**Decision:**

Reject

**Comment:**

AgentRx presents a diagnostic framework for localizing critical failure steps in failed agent trajectories, with a benchmark of 115 human-annotated trajectories and a grounded-theory-derived 9-category failure taxonomy.

Reviewers agreed on the importance and originality of targeting the first unrecoverable failure. Concerns centered on benchmark scale (some subsets as small as 16 trajectories), limited comparison to related work (especially MAST), and modest domain-dependent empirical gains. The rebuttal addressed annotation quality (mean κ = 0.804), added cost analysis, and showed structural generalizability on MAST, but two reviewers maintained weak reject with evaluation scale as an unresolved concern.